# Diffusion on PCA-UMAP Manifold: The Impact of Data Structure Preservation to Denoise High-Dimensional Single-Cell RNA Sequencing Data

**DOI:** 10.3390/biology13070512

**Published:** 2024-07-09

**Authors:** Padron-Manrique Cristian, Vázquez-Jiménez Aarón, Esquivel-Hernandez Diego Armando, Martinez-Lopez Yoscelina Estrella, Neri-Rosario Daniel, Giron-Villalobos David, Mixcoha Edgar, Sánchez-Castañeda Jean Paul, Resendis-Antonio Osbaldo

**Affiliations:** 1Human Systems Biology Laboratory, Instituto Nacional de Medicina Genómica (INMEGEN), Periferico Sur 4809, Arenal Tepepan, Tlalpan, Mexico City 14610, Mexico; padron_cristian@comunidad.unam.mx (P.-M.C.); avazquez@inmegen.gob.mx (V.-J.A.); desquivel@cua.uam.mx (E.-H.D.A.); danielneri08@facmed.unam.mx (N.-R.D.); davidgironvillalobos@comunidad.unam.mx (G.-V.D.); emixcohahe@conahcyt.mx (M.E.);; 2Programa de Doctorado en Ciencias Biomédicas, Circuito Posgrados, Ciudad Universitaria, Alcaldía Coyoacán Unidad de Posgrado Edificio B primer Piso, Universidad Nacional Autónoma de México (UNAM), Mexico City 04510, Mexico; 3Programa de Doctorado en Ciencias Médicas, Odontológicas y de la Salud, Unidad de Posgrado, Edificio A, 1er Piso, Circuito Posgrados, Ciudad Universitaria, Alcaldía Coyoacán, Universidad Nacional Autónoma de México (UNAM), Mexico City 04510, Mexico; 4Programa de Maestría en Ciencias Bioquímicas, Unidad de Posgrado, Edificio B, 1er Piso, Circuito de los Posgrados, Ciudad Universitaria, Universidad Nacional Autónoma de México (UNAM), Alcaldía Coyoacán, Ciudad de México 04510, Mexico; 5CONAHCYT-INMEGEN, Periferico Sur 4809, Arenal Tepepan, Tlalpan, Mexico City 14610, Mexico; 6Coordinación de la Investigación Científica-Red de Apoyo a la Investigación, Instituto Nacional de Ciencias Médicas y Nutrición Salvador Zubirán, Vasco de Quiroga, 14, Belisario Dominguez Sección XVI, Tlalpan, Mexico City 14080, Mexico; 7Centro de Ciencias de la Complejidad, Unversidad Nacional Autónoma de México (UNAM), Circuito Centro Cultural, Coyoacán, Mexico City 04510, Mexico

**Keywords:** manifold learning, UMAP, diffusion maps, scRNA-seq, imputation, denoising, high-dimensional data

## Abstract

**Simple Summary:**

In scRNA-seq analysis, diffusion-based approaches help identify the connections between cells, allowing us to observe the progression of individual cells as they change phenotypes within a mathematical space known as a manifold. Recently, these approaches have been used as a reference for imputation, a technique that addresses missing data, a common challenge in scRNA-seq analysis. For example, MAGIC is a popular diffusion-based imputation method, and it has shown success in uncovering gene–gene interactions related to phenotypic transitions that would not be possible without imputation. However, previous evaluations have not adequately compared the impact of different parameter settings on MAGIC, particularly over-smoothing issues. To address this, we developed sc-PHENIX, which utilizes a similar diffusion approach as MAGIC but incorporates a PCA-UMAP initialization step, whereas MAGIC only uses PCA. We compared sc-PHENIX and MAGIC in terms of imputation accuracy, visualization, biological insights, and preservation of data structure. Our findings show that sc-PHENIX outperforms MAGIC across various common parameters such as “diffusion time” (*t*), the number of nearest neighbors (*knn*), and PCA dimensions. It effectively captures and preserves the global, local, and continuous data structures, leading to more reliable imputation and potentially uncovering new biological insights in diverse datasets.

**Abstract:**

Single-cell transcriptomics (scRNA-seq) is revolutionizing biological research, yet it faces challenges such as inefficient transcript capture and noise. To address these challenges, methods like neighbor averaging or graph diffusion are used. These methods often rely on k-nearest neighbor graphs from low-dimensional manifolds. However, scRNA-seq data suffer from the ‘curse of dimensionality’, leading to the over-smoothing of data when using imputation methods. To overcome this, sc-PHENIX employs a PCA-UMAP diffusion method, which enhances the preservation of data structures and allows for a refined use of PCA dimensions and diffusion parameters (e.g., k-nearest neighbors, exponentiation of the Markov matrix) to minimize noise introduction. This approach enables a more accurate construction of the exponentiated Markov matrix (cell neighborhood graph), surpassing methods like MAGIC. sc-PHENIX significantly mitigates over-smoothing, as validated through various scRNA-seq datasets, demonstrating improved cell phenotype representation. Applied to a multicellular tumor spheroid dataset, sc-PHENIX identified known extreme phenotype states, showcasing its effectiveness. sc-PHENIX is open-source and available for use and modification.

## 1. Introduction

Single-cell RNA sequencing (scRNA-seq) is a revolutionary technology that portrays the transcriptome profiles of thousands of individual cells disaggregated from human tissue or cultured cells in vitro. Thus, it provides a higher biological resolution than bulk RNA sequencing, which is suitable for exploring questions around the structural and functional heterogeneity inside and among biological samples [1]. However, scRNA-seq protocols suffer from various noise sources. 

The most detrimental effect is ‘dropout events’, which comprise the behavior of those genes with discordant gene expression levels along cells of the same cellular type [2]. Consequently, this effect influences the count matrix, which in a typical scRNA-seq experiment has an excess of zeros, and only a tiny fraction of transcripts are detected in each experiment (10–15%) [3,4]. Therefore, due to this inherent stochasticity produced by the low initial concentration of mRNA for individual cells, relationships among gene–gene profiles are lost, and only the most robust correlations prevail [5]. 

Imputation strategies combined with dimensional reduction methods have been suggested to recover the missing expression profiles and systematically reduce the technical noises [6]. These methods have significantly contributed to extracting meaningful biological information and identifying novel cell subpopulations [7]. However, an accurate approximation of low-dimensional embeddings is a difficult task because Euclidean distances among samples tend to be homogeneous in high-dimensional spaces (also known as the “curse of dimensionality”) [8]. Consequently, data structure preservation will be lost, and several samples from distinct single-cell phenotypes could become spurious nearest-neighbors in the low dimensional representation of these methods [8,9].

Even though there is no definition of preserving local or global structures [10], the degree of preservation of the data structures varies depending on the dimensional reduction method [11]. For instance, global structure preservation considers the well-arranged or disposed distinct clusters in space, rendering an overall view of the system [11]. Meanwhile, the local structure reveals fine-grained details within inner clusters and is often seen as a defined separation of distinct groups that provide relevant information on heterogeneity. In the case of gene expression data, the continuum structure is usually described as trajectories or branches, where samples go through a continuum of gradual and progressive changes in transcriptomic single-cell profiles [11]. In this case, a well-balanced global and local structure allows a continuum structure of distinct and connected cell phenotypes.

Researchers have developed several dimensional reduction methods to address the challenges posed by the high dimensionality of scRNA-seq data. Principal Component Analysis (PCA) [12], t-distributed Stochastic Neighbor Embedding (t-SNE) [13], and Uniform Manifold Approximation and Projection (UMAP) [14] are among the key techniques employed to simplify the complexity of datasets while preserving essential biological information. For instance, Principal Component Analysis (PCA), a linear dimensionality reduction method, is often favored for its ability to provide a faithful representation of the global structure of the data [11]. However, PCA partially loses local structure, making distinguishing fine-grained details where distinct gene profiles coexist challenging. 

As a result, the projection of the samples into lower dimensions often overlaps, leading to a noisy visualization of scRNA-seq data [11]. Non-linear dimensionality reduction methods like UMAP and t-SNE effectively capture the fine-grained local structure of scRNA-seq data, yet they may distort the continuum and global structure [11]. Concerning continuum structure preservation, UMAP and t-SNE often shatter trajectories into discrete clusters, falsely suggesting natural clustering in low-dimensional visualizations [11]. Regarding global structure preservation, there is evidence that UMAP has better computational performance than t-SNE by successfully capturing much of the large-scale global structure of the multidimensional data similar to what is well represented by Laplacian Eigenmaps and PCA [14]. 

In addition, to preserve the global data structure, PCA has been used as an initialization step for t-SNE [15]. Similarly, for UMAP visualizations, Sakaue et al. found that the arrangement of clusters is more consistent with biological reality [16]; for instance, in genomic data, similar subgroups are grouped, and phenotypically distinct groups are more distant. Therefore, PCA-UMAP initialization performed better than UMAP, t-SNE, and PCA-t-SNE in preserving the global structure. Additionally, the clusters were arranged similarly to PCA, indicating another sign of global structure preservation [16]. Not just that, PCA initialization for UMAP effectively reveals fine-scale population structures by preserving local details [16]. 

Unfortunately, t-SNE is limited to reducing dimensionality to three or fewer dimensions, resulting in the loss of high-dimensional data information. In contrast, UMAP faces no such restrictions and can more closely match the dimension of the underlying manifold of scRNA-seq data. Additionally, UMAP is computationally faster than t-SNE for large datasets such as sc-RNAseq [14]. 

The embedding space generated by these dimensional reduction methods has been successfully applied in various contexts [5,6,11,15,17,18]; one of its applications is data imputation [6,19]. Concerning scRNA-seq data, imputation methods are designed to recover missing gene expressions in the count matrix, reflecting a more accurate transcriptome scenario. Generally, these methods fall into four categories: model-based, smoothing-based, machine learning-based, and matrix theory-based [6,20]. 

The different scRNA-seq imputation methods have already been benchmarked [6,20]. For example, methods like Markov Affinity-based Graph Imputation of Cells (MAGIC) [5,21], K-Nearest Neighbor Smoothing (*knn*-smoothing) [19], and Single-cell Analysis Via Expression Recovery (SAVER) [22] excel due to their ability to enhance the identification of differentially expressed genes. These methods also improve unsupervised clustering, facilitate trajectory analyses, and handle memory usage and scalability efficiently, as demonstrated in various comparative evaluations [6,19,22]. 

Although SAVER, *knn*-smoothing, and MAGIC can recover lost data in many cases, a drawback of these methodologies is the false association commonly known as “over-smoothing”, which can result in inaccurate gene relationships and inadequate biological insights [23]. For example, MAGIC imputes by sharing information across similar cells via data diffusion on a PCA space. Hence, it is based on the diffusion map theory [5,21]. Nevertheless, assessments of MAGIC have revealed potential drawbacks, including the distortion of gene expression by conflating distinct phenotypes and inaccurate recovery of specific markers [24]; these findings underscore the risk of over-smoothing data. Indeed, inadequate management of over-smoothing results in failure to enhance their performance in downstream analyses compared to non-imputation approaches [6]. However, there are imputation methods designed to address over-smoothing and preserve biological zeros, such as ALRA [24]. 

Linderman et al. showed that MAGIC tends to over-smooth data, whereas ALRA outperforms it in terms of imputation. However, that article lacks a discussion on tuning different parameters for MAGIC, as they used MAGIC with default parameters. This is a recurrent problem where new methods appear and create their own benchmarking. Now, in terms of scRNA-seq imputation benchmarkings, [6,20] showed that MAGIC outperforms several imputation methods, including ALRA. Hou et al. showed that MAGIC achieves equilibrium for downstream analyses but was well suited for data structure preservation [6].

On the other hand, Cheng et al., when using default parameters, showed that MAGIC-imputed values were much smaller compared to the estimated standard deviations from other imputation methods [25]. Additionally, analyzing gene–gene correlations, like in the case of MAGIC, should be treated with caution. 

In this context, the necessity to develop novel imputation methods that amalgamate the advantages of state-of-the-art techniques and enhance gene expression imputation performance is imperative. To address this issue, we introduce *single cell-PHEnotype recovery by Non-linear Imputation of gene eXpression* (sc-PHENIX) as a contribution to this evolving field. sc-PHENIX is an open-source Python package that implements a hybrid unsupervised machine learning approach for recovering missing gene expression in single-cell data (Figure 1). 

In summary, our proposed method builds upon the principles of MAGIC, incorporating a crucial initial step to mitigate over-smoothing. This step leverages the inherent data structure present in high-dimensional spaces through PCA-UMAP initialization, enhancing the performance of the method (Figure 1A). 

Specifically, sc-PHENIX resolves the challenges stemming from diffusion on PCA space, as seen in MAGIC, by using diffusion on PCA-UMAP space. Our results demonstrate that sc-PHENIX effectively approximates the underlying cellular structure, leading to accurate recovery of gene expression patterns among similar cells. Notably, our method unveils previously hidden biological insights, surpassing the performance of current state-of-the-art imputation methods. sc-PHENIX is open-source, available for use at https://github.com/resendislab/sc-PHENIX, accessed on 23 February 2024.

## 2. Materials and Methods

### 2.1. Preprocessing of the Row Count Single-Cell Matrix

We conducted quality control on the data before applying an imputation method, such as MAGIC or sc-PHENIX. This includes filtering steps, such as removing low-quality cells, discarding empty droplets in droplet-based technologies, or eliminating empty genes. Subsequently, we applied a library size normalization to obtain the scRNA-seq input data for the imputation method. We described the preprocessing steps for each one of the used datasets in Section 2.6 and Section 2.7. Notably, applying library size normalization tailored to the specific technology reduces biases related to cell size in measurements. 

This normalization ensures that the resulting weighted neighborhood construction remains unbiased [5]. The input for the sc-PHENIX pipeline is either the raw or preprocessed expression matrix (***D***) (Figure 1A). Lytal et al. suggest that library size normalization is particularly well-suited for scRNA-seq data; however, we provide the flexibility for end-users to choose the appropriate normalization method [26]. Additionally, if users apply the MAGIC method, we recommend employing the normalization approach described by Van Dijk et al. [5]. 

### 2.2. PCA-UMAP Manifold and Creation of the Distance Cells Matrix

All steps with the implementation of equations and generated matrices used in sc-PHENIX are depicted in Figure 1B for the computation of the ***M*^t^** and imputation. The initial step entails the precise construction of ***D_Dist_*** from the ***D*** matrix, which is pivotal for the success of sc-PHENIX in avoiding over-smoothing in scRNA-seq data via diffusion on manifolds (Figure 1A). To this end, the preprocessed single-cell count matrix ***D*** is dimensionally reduced using PCA as an informative initialization step for UMAP, resulting in the creation of the PCA-UMAP space matrix (***D*_PCA-UMAP_**). 

UMAP embeddings represent projections that can be interpreted as Euclidean distances, making them suitable for constructing a Euclidean distance matrix (***D*_Dist_**) [14]. The primary distinction between sc-PHENIX and MAGIC is the use of PCA-UMAP space to compute the ***D*_Dist_** of cells instead of solely relying on PCA space like MAGIC. Additionally, sc-PHENIX offers users the flexibility to employ any manifold initializations.

As previously reported, the PCA-UMAP manifold captures local structures, particularly for noisy and high-dimensional data [15]. Thus, PCA serves as a critical initialization step for UMAP, preserving both global [15] and local [16] data structures, thereby accurately representing the scRNA-seq manifold in the PCA-UMAP space. This lower-dimensional representation is crucial for sc-PHENIX’s enhanced imputation performance. However, caution is advised against using very low-dimensional UMAP representations (2D or 3D) as they may fail to fully preserve high-dimensional distances, especially for scRNA-seq data [27]. 

We confirm and discuss this observation in Section 3.2, where imputing high-dimensional bulk data with 80% artificial zeros using more than *n_components* = 3 (UMAP dimensions) yields a higher R2 (coefficient correlation) between the imputed and original data. The strength of UMAP is its adaptability in selecting embedding dimensions, for instance, three components or more, that align with the dimensions of the underlying scRNA-seq data manifold, therefore enhancing clustering. This characteristic is advantageous for density-based clustering or machine learning methods [14].

### 2.3. Creation of the Affinity Matrix

sc-PHENIX uses an adaptive Gaussian kernel to create the affinity matrix (***A*_non-sim_**). The advantages of using an adaptive kernel are already known for the stability of the imputation [5]. We use the same adaptive kernel as MAGIC does, defined in Equation (1).
(1)$$A_{non-sim}\left(i,j\right)=e^{-{\left(\frac{D_{Dist}\left(i,j\right)}{\sigma \left(i\right)}\right)}^{decay}}$$
(2)$$\mathrm{where}\;\sigma \left(i\right)=distance\left(i,neighbor\left(i,knn\right)\right)$$

The effectiveness of this kernel stems from the adaptability of its bandwidth relative to a cell’s neighbors; specifically, the bandwidth $\sigma \left(i\right)$ for a cell i is defined as the distance to its $k_{a}$-th nearest neighbor (*knn*), as shown in Equation (2). The method’s effectiveness comes from adjusting its “bandwidth” based on the proximity of a cell to its neighbors. 

Specifically, the bandwidth for a cell is defined as the distance to its k-th nearest neighbor, allowing the measurement to adapt to the local cell density. Applying this adaptive kernel to the distance matrix (***D*_Dist_**) produces a non-symmetric affinity matrix (***A*_non-sim_**), where the element ***A*_non-sim_(**i, j**)** is not necessarily equal to ***A*_non-sim_(**j, i**)**. We need to obtain an ***A*_sim_** symmetric; this is required as described in Equation (3).
(3)$$A_{sim}=A_{non-sim}+A_{non-sim}^{T}\;$$

### 2.4. Row-Stochastic Markov-Normalization of A into Markov Matrix M

A row normalization of ***A*_sim_** is needed to create the Markov transition matrix ***M***. Each row of this Markov transition matrix must sum 1, showing the probability distribution of the transition from one cell to every other cell in the scRNA-seq data. Row normalization is calculated by dividing each entry in ***A*_sim_** by the sum of *k* rows affinities, Equation (4).
(4)$$M\left(i,j\right)=\frac{A_{sim}\left(i,j\right)}{\sum_{k}A_{sim}\left(i,k\right)}\;$$

### 2.5. Data Diffusion and Imputation

Once ***M*** is calculated, the diffusion process starts with the exponentiation of ***M***; this means that phenotypically similar cells should have strongly weighted affinities, and noise/spurious neighbors are down-weighted in this diffusion process. Raising the ***M*** to the power *t* generates this ${M^{t}}_{\left(i,j\right)}$, which represents the probability that a random walk of length *t* starting at cell *i* will reach cell *j*, same as MAGIC, we call the parameter *t*, the ‘diffusion time’.

As mentioned in van Dijk et al. [5], the Markov affinity matrix ***M*** represents the probability distribution of transitioning from one cell to another (the probability of cell “*i*” and cell “*j*” are similar). After constructing ***M***, data diffusion is achieved through the exponentiation *t* of ***M***. The effect of *t* exponentiation filters out noise and increases the similarity based on strong trends in the data. Thus, the weight (probability) among phenotypically similar cells increases. In contrast, phenotypically distinct cells are down-weighted. 

This noise is presented by technical errors such as dropouts, artifacts, and a small and randomized ratio of the capture of transcripts, all contributing to this noise. Noise does not have a strong trend in the manifold, but similar cells do, and because of that, they are filtered by the exponentiation of the Markov matrix. As *t* increases, diffusion methods learn the manifold structure and remove the noise dimensions. Finally, imputation is a matrix multiplication of the ***M*^t^** and ***D***, Equation (5).
(5)$$D_{imputed}=M^{t}*D$$

In this study, we demonstrate that constructing the ***M*^t^** using PCA initialization fails to adequately separate distinct cell phenotypes, limiting its utility in imputation to sharing information with only a few local neighbors. This results in further data over-smoothing. Over-smoothing, associated with higher values of parameters (*t*, *knn*, or PCA dimensions), can be avoided by restricting information sharing to local cells (local imputation); however, this approach may not recover significant biological insights. For instance, gene–gene interactions, as retrieved by MAGIC in the foundational work by van Dijk et al. [5], are inadequately structured and fail to elucidate the dynamics of gene expression markers across cell states and local granularity. The initial gene–gene interactions identified with lower *t* steps exemplify this issue [5].

In contrast, constructing the ***M*^t^** based on a PCA-UMAP initialization (also known as sc-PHENIX) effectively separates distinct cell phenotypes, facilitating comprehensive cell graph imputation. This approach, detailed in Equation (5), permits sharing information to a more local setting using higher values of parameters (*t*, *knn*, or PCA dimensions), avoiding over-smoothing more effectively. Consequently, it reveals local details of cell types that neither non-imputation methods nor MAGIC can uncover.

### 2.6. Worm Bulk Microarray Data Processing 

To assess the imputation accuracy of MAGIC and sc-PHENIX, we developed a validation dataset derived from bulk transcriptomic data from 206 synchronized *C. elegans* young adults. These measurements were taken at regular intervals over a 12 h developmental period using microarrays [28]. Due to its gradual changes, these data have a continuum structure.

Adapting the approach outlined in the MAGIC paper, we introduced some changes. Specifically, we replaced their method with randomly and uniformly inserted zeros. We exponentiated the log-scaled expression levels matrix and downsampled each entry to retain 80% of the original data. Introducing zeros into the count matrix also causes samples to cluster more, making the computation of similarity more complicated to challenge both imputation methods. We decided to use this corrupted count matrix after exponentiation because the matrix before introducing zeros retains the developmental time order and preserves the continuum structure [5]. 

In scRNA-seq, dropout occurs randomly, meaning any transcript has an equal chance of being missed. Although dropout is more likely with lower gene expression, even highly expressed genes might not be detected in all cells. We expect that the remaining gene expression data will still have enough information to accurately compute similarities for imputation.

### 2.7. scRNA-seq Datasets and Preprocessing

#### 2.7.1. Peripheral Blood Mononuclear Cells Dataset 

**Description**: 3k PBMCs (peripheral blood mononuclear cells) from a healthy donor; PBMCs are primary cells with relatively small amounts of RNA (∼1 pg RNA/cell); 2700 cells were detected. 

**Source**: https://support.10xgenomics.com/single-cell-gene-expression/datasets/1.1.0/pbmc3k, accessed on 4 July 2021.

**Preprocessing**: This dataset is from the tutorial from Seurat; the R vignette can be consulted for quality control, normalization (libsize), and transformation (Log +1) in https://satijalab.org/seurat/articles/pbmc3k_tutorial.html, accessed on 4 July 2021. Additionally, we imputed data using SAVER with default parameters, and for *knn*-smoothing, we varied the values for PCA and *knn* using the preprocessed matrix.

#### 2.7.2. Neuronal Dataset

**Description**: In this dataset published in *Nature Neuroscience*, the researchers constructed a cellular taxonomy of the primary visual cortex in adult mice using single-cell RNA sequencing. They identified 49 transcriptomic cell types, including 23 GABAergic, 19 glutamatergic, and 7 non-neuronal types. The dataset consists of 24,057 genes and 1679 cells.

**Source**: https://singlecell.broadinstitute.org/single_cell/study/SCP6/a-transcriptomic-taxonomy-of-adult-mouse-visual-cortex-visp#study-download (accessed on 4 June 2024).

**Preprocessing**: The normalization they used was RPKM. The imputation parameters were changed based on the combination of each result *t*, *knn*, and PCA dimensions. Other parameters used for MAGIC were default. For sc-PHENIX, the UMAP parameters were as follows: *n_components* = 5, *n_neighbors* = 14, *verbose* = True, *metric* = ‘cosine’, *min_dist* = 0.2, *n_epochs* = 1000, *negative_sample_rate* = 5.

**Visualization of the non-imputed data**: To visualize the imputed data, we used a 2D-UMAP visualization created from the non-imputed data to observe the behavior of imputed single cells in this approximation. For that, we created the 2D-UMAP for the non-imputed data where the umap parameters were as follows: *n_components* = 2, *n_neighbors* = 10, *verbose*= True, *metric* = ‘cosine’, *min_dist* = 0.5, *n_epochs* = 1500, *negative_sample_rate* = 5, *set_op_mix_ratio* = 0.8, *learning_rate* = 0.1, *local_connectivity* = 3, *random_state* = 42. The input was the RPKM normalized data. Here, we imputed this dataset with knn-smoothing parameters based on different value combinations of PCA and *knn*.

#### 2.7.3. Epithelium Mesenchyme Transition Dataset

**Description**: In the Epithelium mesenchyme transition (EMT) dataset, the authors document single-cell RNA sequencing with the inDrop method to analyze an HMLE breast cancer cell line undergoing EMT. The study captured data from 7523 single cells and 28,910 genes at 8 and 10 days after stimulation with TGF-beta. The authors derived this dataset from the original MAGIC publication. We chose this dataset to evaluate the reproducibility of sc-PHENIX compared to MAGIC when sc-PHENIX utilizes PCA for initialization.

**Source**: https://github.com/KrishnaswamyLab/MAGIC/blob/master/data/HMLE_TGFb_day_8_10.csv.gz (accessed on 4 June 2024), GEO Accession: GSE114397.

**Preprocessing**: We used the same normalization method described in the published MAGIC article, employing the scprep library. The code lines used were scprep.normalize.library_size_normalize() and scprep.filter.filter_library_size (*cutoff* = 1500). These parameters were the same as those used in the published article. scprep is a comprehensive Python framework designed for loading, preprocessing, and visualizing data, particularly in single-cell genomics. (https://scprep.readthedocs.io/en/stable/reference.html), accessed on 4 June 2024. The imputation parameters were *t* = 6, *knn* = 30, *decay* = 15, and the first 20 PCA dimensions. 

#### 2.7.4. Tumor Spheroid Dataset

**Description**: The tumor spheroid dataset contains 364 cells and 23,922 genes. The raw data and the raw count matrix are available through the Gene Expression Omnibus (GEO) with accession number GSE145633. The data are the same as in https://github.com/resendislab/sc-PHENIX, accessed on 4 June 2024. 

**Limitation with MAGIC**: One limitation of MAGIC is that it can create a low-dimensional manifold for initializing the embedding when gene filtration (most variable genes) is performed, potentially filtering out significant genes during imputation. MAGIC recommends creating a manifold embedding using the most variable genes. Cellular subpopulations with unique expression profiles can be more effectively identified by utilizing variable genes, leading to a better understanding of cellular heterogeneity. This information can be valuable for constructing meaningful low-dimensional manifolds. Pre-filtering data by the most variable genes assumes that genuine biological differences will manifest as increased variation in the affected genes, compared to other genes that are only affected by technical noise or a baseline level of “uninteresting” biological variation or a baseline level of “uninteresting” biological variation (e.g., from transcriptional bursting). 

However, this has a side effect; for example, with MAGIC, during the filtration of genes with *percentile* = 85%, genes such as *VIM* marker (vimentin) do not appear in the recovered dataset among more than half of the genes. However, you should note that the filtration step for the most variable genes may cause the loss of most transcription factors and membrane receptor transcripts due to their low abundance.

**Opportunities with sc-PHENIX**: In sc-PHENIX, we can recover all genes and create a PCA-UMAP initialization from the most variable genes, allowing it to receive these inputs. In the MAGIC article, van Dijk et al. utilized all genes to recover expression for the EMT dataset [5]. This approach allows us to evaluate gene–gene interactions, dropout events, and filtration steps, such as filtering for the most variable genes, which would typically mask these interactions (https://github.com/dpeerlab/magic/blob/published/notebooks/Magic_single_cell_RNAseq.ipynb, accessed on 4 June 2024).

**Preprocessing**: For MAGIC’s imputation, the initial data were preprocessed using all 23,922 genes to create the preprocessed count matrix using the library_size_normalize() function from the scprep library. After that, we computed the square root of the data. Based on the preprocessed count matrix, the parameters for MAGIC were *t* = 5 and *knn* = 20. MAGIC used 100 PCA dimensions directly from the preprocessed count matrix. 

For sc-PHENIX’s imputation, the preprocessed count matrix was created by initially using the library_size_normalize() function from the scprep library. After that, we computed the square root of the data. For PCA-UMAP initialization, we used the already normalized and transformed count matrix. Subsequently, we identified the most variable genes using scprep function scprep.select.highly_variable_genes(data_sqrt, *percentile* = 85). Finally, we transformed the data to 30 PCA dimensions and then to 30 UMAP dimensions. For UMAP, the parameters were as follows: *n_components* = 30, *metric* = ‘euclidean’, *n_epochs* = 1000, *min_dist* = 0.5, *verbose* = 1, *random_state* = 1. Imputation parameters for sc-PHENIX were *t* = 5, *decay* = 5, and *knn* = 20.

**Characterization of phenotypes MCF7 clusters**: After preprocessing the data, we compared the imputation results of MAGIC and sc-PHENIX across all genes. To investigate extreme phenotypes within our sc-RNAseq data, we embedded the imputed data into a 3D PCA plot, a similar approach to van Dijk et al. [5], which helps visualize structural patterns. Archetypal analysis, introduced by Adele Cutler and Leo Breiman in 1994, focuses on ‘archetypes’, which are extremal points that are convex combinations of observations [29]. These archetypes visually manifest as points at the extremities of the plot’s data distribution, highlighting the dataset’s most distinctive and divergent phenotypes.

In this 3D space, we applied Hierarchical Density-based spatial clustering of applications with noise (HDBSCAN) [30] to the PCA projections. HDBSCAN identifies dense clusters, enhancing our analysis by isolating robust clusters near the archetypes and filtering out “noise”, HDBSCAN noise clusters are labed as −1.

By integrating 3D PCA with HDBSCAN, we established a framework that aids data visualization and enhances our ability to analyze cellular states at the extremes. This approach enabled us to capture and interpret critical phenotypic expressions, providing a deeper understanding of the biological processes involved. Finally, we visually identified extreme archetype clusters as those located on the edges of the geometric manifold of the 3D PCA plot.

**Differential expression by clusters**: To quantify the differences in gene expression across clusters for the imputed data with sc-PHENIX and MAGIC, we used the Earth Mover’s Distance (EMD) metric for differential gene expression by clusters, using the approach of one vs. all. EMD, also known as the first Wasserstein distance, which measures the minimal cost of transforming one probability distribution into another by estimating the required changes to shape one distribution into another. The EMD score, multiplied by the sign of the mean gene difference, denotes the overall direction of the gene expression shift. 

The EMD between two distributions *u* and *v* is calculated as follows:(6)$$l_{1}(u,v)=\int_{-\infty }^{\infty }|U(x)-V(x)\;|\;dx$$
where *U(x)* and *V(x)* are the cumulative distribution functions of *u* and *v*, respectively.

In our study, EMD was used to compare the gene expression profiles of each cluster versus the rest of the samples. To denote the direction of gene expression changes, the EMD score is multiplied by the sign of the mean difference in expression between the cluster and other samples:(7)$$EMD\;score=EMD\times sign\;({\bar{x}}_{cluster}-{\bar{x}}_{rest})$$

The term sign $\left({\bar{x}}_{cluster}-{\bar{x}}_{rest}\right)$ is a mathematical function that returns the sign of the difference between the mean gene expression in a specific cluster $\left({\bar{x}}_{cluster}\right)$ and the mean gene expression in the rest of the samples $\left({\bar{x}}_{rest}\right)$. This function returns +1 if the difference is positive, indicating that the mean expression in the cluster is higher than in the rest. 

The function returns −1 if the difference is negative, and it returns 0 if there is no difference. This adjusted EMD score helps identify the magnitude and direction of gene expression shifts, providing insights into whether genes are upregulated or downregulated within specific clusters. We used the scprep library to perform all these steps to calculate the EMD score.

**Gene set enrichment analysis of clusters**: We used WebGestalt (WEB-based Gene SeT AnaLysis Toolkit) to perform the enrichment, as used in other biological models and data [31,32,33,34,35]. We considered only the HALLMARKS pathways of Molecular Signatures Database (MSigDB) v7.5.1 to facilitate the analysis. The EMD score values were used with WebGestalt (http://www.webgestalt.org/), accessed on 1 June 2023 [31]; the functional database was obtained from MSigDB gene sets (http://www.gsea-msigdb.org/gsea/msigdb/collections.jsp#H, accessed on 23 February 2024) in the h.all.v7.5.1.symbols.gmt file (hallmark gene sets) and c2.cp.reactome.v2022.1.Hs.symbols.gmt file (REACTOME gene sets). WebGestalt has the freedom to use several statistical tests, is versatile within several databases and enrichment pipelines, and is user-friendly. Instead of collapsing data by the mean of gene cluster expression, we used the EMD score values. 

The reason is that the clusters in scRNA-seq data can have multiple distributions that can bias the Gene Set Enrichment Analysis (GSEA) results. Using the mean of the imputed data will recover several biased enriched pathways [36]. EMD does not have the assumption of distributions; more details of the EMD in scRNA-seq data [5]. The statistical significance of enriched pathways was set to an FDR < 0.05. The enrichment statistic parameter used was *p* = 0. We used the statistically significant pathway with their respective normalized enrichment score (NES) values for this analysis.

### 2.8. Multidimensional Scaling of Exponentiated Transition Markov Matrix

The PCA space and ***M*^t^** (from PCA and PCA-UMAP initialization) are transformed into a distance matrix, ($D_{Dist}$) to apply multidimensional scaling (Figure 1C). For faster optimization and more accurate preservation of the high-dimensionality distances in the low-dimensional manifold, we used the algorithm in this multidimensional scaling approach [37] to minimize its energy function, known as stress, by using stochastic gradient descent to move a single pair of vertices at a time. Their results show that stochastic gradient descent can reach lower stress levels faster and more consistently than majorization without needing help from good initialization. The approach proposed in [37] can project data from a distance matrix into 2D and 3D (MDS dimensions). MAGIC returns only the imputed matrix, not its ***M*^t^**. For that purpose, the ***M^t^*** of MAGIC indirectly calculates the ***M*^t^** of MAGIC by computing the ***M*^t^** with sc-PHENIX with PCA initialization. (See Appendix A for further details about the reproducibility of sc-PHENIX as MAGIC with initialization PCA space). 

For the neuronal dataset, we created several 2D plots from the ***M*^t^** of MAGIC and sc-PHENIX based on different combinations of PCA dimensions, *t*, and *knn* values. This was carried out to evaluate the performance of cluster structures. We used a *knn* classifier from the *scikit-learn* library, KNeighborsClassifier [38], with *knn* = 30 to assess local structure preservation. Using the cluster labels from the 2D MDS embedding, we measured the accuracy of the *knn* classifier in predicting these 21 phenotype labels. This approach helps determine how well the local cluster structures are preserved in the 2D projections from MAGIC and sc-PHENIX.

### 2.9. PHATE Visualization

To evaluate our visualization method, we compared it with the Potential of Heat Diffusion for Affinity-based Transition Embedding (PHATE) [11]. PHATE is designed to preserve local, global, and continuum structures in data using a unique informational distance metric for low-dimensional embeddings. It incorporates PCA in preprocessing to enhance robustness and reliability [11]. We used the MNIST dataset with *knn* = 5, *n_pca* = 500, *n_components* = 2 and the Neuronal dataset with *knn* = 30, *n_pca* = 500, *n_components* = 3 and 2.

### 2.10. Imputation Performance of Neuronal Cells

We evaluated the performance of MAGIC and sc-PHENIX using different combinations of diffusion parameters and PCA dimensions on the neuronal dataset from the adult mouse visual cortex, as provided by Tasic et al. [39]. This neuronal dataset includes 21 well-characterized cell phenotypes. We chose this particular dataset because previous studies, such as Mukherjee et al. [40], have shown that MAGIC’s imputation tends to group several phenotypes, making it a challenging dataset for evaluating the imputation performance of MAGIC and sc-PHENIX. We used a post hoc test called Tukey’s Test [41], which allows us to make pairwise comparisons between the means of each group while controlling for the family-wise error rate (FWRD). 

The differential expression for the imputed gene markers was computed using Tukey’s HSD (honestly significant difference) with an FWRD = 0.05. We used a Python function named pairwise_tukeyhsd from the library statsmodels.stats.multicomp (https://github.com/statsmodels/statsmodels, accessed on 1 June 2023). We considered the use of Tukey’s HSD for post-imputation data evaluation because it is particularly suitable for scRNA-seq datasets imputed by diffusion methods such as MAGIC and sc-PHENIX. These methods tend to smooth gene expressions and thereby approximate normality (more homogeneous) in the data of neighboring samples. Without imputation, using Tukey’s HSD would not be appropriate. The intention of this evaluation is to observe if imputation spreads gene expression among phenotypically distant cells.

We evaluated *Flt1* (NonNeu_Endo and NonNeu_SMC cell types), *Chat* (GABA_Vip cell type), *Sst* (GABA_Sst cell type), and *Serpin11* (Gluta_L6B cell type) gene markers from the imputed data by MAGIC and sc-PHENIX. These markers used as a reference for the quality of imputation were determined by the dataset from the work of Tasic et al. We created a confusion matrix to represent the expression status of selected gene markers. In this matrix, a value of 1 denotes that the gene marker is differentially expressed in a phenotype, while a value of 0 signifies no differential expression. 

Each confusion matrix is from imputation from MAGIC and sc-PHENIX using a combination of parameters, PCA dimensions, *t*, and *knn*; we obtained precision, recall, and f1-score metrics, and more values of these metrics indicate better quality imputation, meaning a more accurate preservation of gene expression in their respective cell phenotypes. It is worth mentioning that this kind of over-smoothing evaluation needs to be included in scRNA-seq imputation benchmarking [6,20]. Benchmarking focuses on obtaining more recovered differential genes for a good imputation approach. However, the effect of over-smoothing of gene-specific markers post-imputation is not contemplated as quality control. In Appendix A, we provide a visual diagram of the imputation performance.

## 3. Results

### 3.1. Results Overview

The results section of this study is organized into several subsections that systematically evaluate the performance and characteristics of the sc-PHENIX method compared to MAGIC. Each subsection addresses different aspects of the data analysis and imputation process, providing a comprehensive view of how sc-PHENIX outperforms or complements existing methods. For the sake of simplicity, we divided the discussion into the following sections:
**Performance of Diffusion-Based Imputation (Section 3.2)**: The impact of PCA and PCA-UMAP initialization on imputation performance was evaluated using bulk transcriptomic data. Various parameter settings, such as *t*, *knn*, and PCA dimensions, are used to assess the imputation performance.**Visualization with sc-PHENIX (Section 3.3)**: Various visualizations are provided to demonstrate how sc-PHENIX minimizes over-smoothing compared to MAGIC. This section includes detailed analyses using different datasets, such as MNIST and neuronal datasets, to show the preservation of local and global data structures. Additionally, for the PBMC dataset, we visually evaluated the effect on gene–gene interactions.**Evaluation of Over-Smoothing (Section 3.4):** Over-smoothing is analyzed by evaluating specific gene markers across different cell phenotypes in the neuronal dataset. The ability of sc-PHENIX to maintain the integrity of these markers without excessive smoothing is compared to MAGIC. **Evaluation of the Heterogeneity of MCF7 Cells Data (Section 3.5):** The heterogeneity of spheroid data from MCF7 breast cancer cells is examined post-imputation. Differences in 3D PCA manifolds and the identification of dense clusters are discussed. The analysis highlights how sc-PHENIX captures more transition states and extreme phenotypes compared to MAGIC.

### 3.2. Performance of Diffusion-Based Imputation Using Bulk Data: Evaluating the Effects of PCA and PCA-UMAP Initialization

To illustrate the effect of using PCA initialization, as implemented in MAGIC, compared with PCA-UMAP initialization, as proposed in this work, on imputation performance, we utilized bulk transcriptomic data from 206 developmentally synchronized young adult *C. elegans*. We introduced random zeros to 80% of these data to corrupt them. The dataset includes bulk RNA-seq samples representing developmental trajectories, measured at regular intervals throughout a 12 h developmental period using microarrays.

We evaluated two scenarios with PCA: high *knn* value (*knn* = 30) and low *knn* value (*knn* = 5). In Figure 2A (high *knn*), 2B (low *knn* for MAGIC, high *knn* for sc-PHENIX), and 2C (low *knn*), we fixed *t* = 5 and *decay* = 15. In Figure 2A, which represents the high *knn* scenario, the graphs display Pearson Correlation, Spearman correlation, and R^2^ scores as a function of the number of PCA components. The orange lines represent sc-PHENIX’s performance, while the blue lines represent MAGIC’s performance. The shaded areas around the lines indicate the confidence interval (95%). We observed that sc-PHENIX consistently shows higher values across all metrics compared to MAGIC, indicating better similarity to the original data.

In Figure 2C, representing the low *knn* scenario, MAGIC performs better initially with fewer PCA components (approximately more than 10 but less than 25), indicating that it is more effective in this scenario at the beginning. However, its performance decreases as the number of PCA components increases. On the other hand, sc-PHENIX initially performs lower with fewer than approximately 100 PCA dimensions, not surpassing MAGIC with fewer PCA dimensions. Although sc-PHENIX’s performance improves as the number of PCA components increases, it does not surpass MAGIC’s performance with few PCA dimensions. Overall, in the low *knn* scenario, the area under the curve (AUC) metrics (see, Spearman and Pearson) shows little difference, indicating that there is not much difference between the methods in terms of performance robustness through PCA dimensionality. 

Figure 2A,C show that MAGIC works better with low *knn* values, while sc-PHENIX performs better with higher *knn* values. To make a fair comparison between both methods, we visualized the scenario of low *knn* for MAGIC and high *knn* for sc-PHENIX in Figure 2B. We observed that MAGIC’s performance decreases significantly, whereas sc-PHENIX maintains higher performance, highlighting its robustness with high *knn* values as PCA dimensionality increases.

Additionally, we used the AUC to evaluate its robustness along the PCA dimensionality of both methods. The AUC values for sc-PHENIX were consistently higher than MAGIC, demonstrating a better capability to recover the underlying structure of the original data. Specifically in Pearson metrics, the AUC values were as follows: for the high *knn* scenario (Figure 2A), sc-PHENIX achieved an AUC of 0.76, while MAGIC achieved an AUC of 0.68; for the low *knn* scenario (Figure 2C), sc-PHENIX achieved an AUC of 0.71, while MAGIC achieved an AUC of 0.72; and for the mixed *knn* scenario (Figure 2B), sc-PHENIX maintained an AUC of 0.76, while MAGIC’s AUC was 0.72.

Thus, based on the AUC metric in Figure 2A–C, MAGIC is more effective with fewer PCA components and a low *knn* value. In contrast, sc-PHENIX performs better with more PCA components and requires a higher *knn* value to achieve superior imputation performance, generally outperforming MAGIC, regardless of whether MAGIC uses high or low *knn* values. These results suggest that sc-PHENIX’s initialization method shows more robustness and scalability across various PCA dimensions when values of *knn* are high.

We demonstrated the implications of PCA-UMAP initialization versus UMAP-only initialization for sc-PHENIX imputation. We fixed the parameters at *t* = 5, *decay* = 5, *n_pca* = 71, and *knn* = 30. Our results showed that for this dataset, using PCA as an initialization for UMAP results in better performance for sc-PHENIX imputation compared to using UMAP alone with any of the three metrics (Figure 2D). Furthermore, we found that more UMAP dimensions lead to better performance, with at least three UMAP dimensions maintaining robustness as dimensionality increases. This robustness is likely because the dataset is small and has a relatively uncomplicated topological structure. In Figure 2E, we observed that using a 2D UMAP plot of developmental time samples aligns well with the manifold continuously, without branching, indicating a uniform developmental trajectory without bifurcations.

In Figure 2H, we show that 71 PCA components achieve 70% of the variance of dropout data. Based on this, in Figure 2E, we set *n_pca* to 71 according to MAGIC’s recommendation to capture at least 70% of the data’s variability. However, the performance of MAGIC’s imputation was negatively affected by using more than 71 PCA dimensions, leading to over-smoothing. In contrast, we found that PCA-UMAP for sc-PHENIX mitigates this issue. These observations suggest that the dataset’s topology is simple and can be approximated with a few PCA components. However, in real applications using scRNA-seq data, there is no reference for selecting the number of PCA dimensions different from the captured variance. This analysis shows that this limitation does not apply to sc-PHENIX due to its robustness to PCA dimensionality.

We observed the order of developmental time along the gene expression of three genes—*C27A7.6, dct-5, and erd-2*—which were selected for their particular non-linear trends (Figure 2F). Additionally, we compared 2D UMAP plots from the original data and we examined the expression of the imputed, corrupted, and original data on the same UMAP. We fixed the parameters for imputation at *t* = 5, *decay* = 5, *n_pca* = 71, *knn* = 30 for sc-PHENIX, and *knn* = 5 for MAGIC. We found that sc-PHENIX localizes gene expression among local *knn* samples with similar expression levels in the reference and does not over-smooth the data as MAGIC does. This sc-PHENIX feature is essential for preserving low or near-zero values, as both biological zeros and technical error zeros are significant.

We plotted samples along developmental time using the same genes and compared increasing *knn* values (Figure 2F). We observed the real values, zero-induced points, and imputations from sc-PHENIX and MAGIC using different *knn* parameter values. sc-PHENIX mitigates over-smoothing, whereas MAGIC tends to over-smooth as *knn* values increase. As *knn* values increase for MAGIC, gene expression trends average out due to its diffusion-like behavior amplified by its *knn* closeness, but sc-PHENIX effectively mitigates this issue. However, although sc-PHENIX seems to maintain the general trend, it starts losing its ability to retain low-value expression due to MAGIC’s diffusion-like behavior in high *knn* values (30 and 40). However, the general trend is maintained because sc-PHENIX accurately separates and groups samples.

There is the question of whether t-SNE performs better than UMAP. We put it to the test and found that neither PCA-tSNE nor t-SNE initialization outperformed sc-PHENIX with PCA-UMAP or UMAP initialization, as shown in Figure 2G. It is important to note that t-SNE can only reduce dimensionality to three dimensions. When using one-dimensional PCA-tSNE initialization, it performed better than two tSNE components. However, using t-SNE initialization compared to UMAP or PCA-UMAP initialization was worse. 

We went further and analyzed another important initialization for UMAP, which is Laplacian Eigenmaps. Kobak et al. showed that using it in conjunction with UMAP generates more accurate embeddings [15]. We tested this hypothesis in Appendix A, where we examined both low and high values of ‘*n_components*’ for Laplacian Eigenmaps across increasing values of *knn*. However, in any case, Laplacian Eigenmaps-UMAP initialization showed a lower AUC and global mean of the metrics—Pearson Correlation, Spearman Score, and R2 Score—compared to any combination of PCA, UMAP, or PCA-UMAP initialization for sc-PHENIX (Figure 2).

Another important parameter that is not further evaluated in MAGIC’s paper is the *decay* parameter. Originally, in the MAGIC article, the formula for the adaptive kernel’s *decay* parameter was set to 2, but this parameter can be tuned. By default, it is set to 1, although the reasons for this are not explained. We evaluated the impact of varying the *decay* parameter alongside increasing PCA dimensions. As shown in Figure 3, MAGIC and sc-PHENIX there is a slight improvement in AUC by 0.01 when the *decay* parameter is increased if we compare it with a fixed *decay* = 5 (Figure 2B best case scenario). Additionally, there is a clearer separation between MAGIC and sc-PHENIX through increasing PCA and *decay* values, with sc-PHENIX demonstrating an overall improvement (see Figure 3).

### 3.3. Visualization with sc-PHENIX

To visually demonstrate how our approach minimizes over-smoothing compared to MAGIC, this section evaluates the cell neighborhoods embedded in the exponentiated Markov matrix (***M*^t^**) for both sc-PHENIX and MAGIC. Additionally, we visualized the imputed data from MAGIC and sc-PHENIX in terms of gene–gene interactions. We have organized the details of the sc-PHENIX implementations into the following subsections:

#### 3.3.1. Visualization of the Exponentiated Markov Matrix Based on Different Manifold Initializations

In sc-PHENIX, or any general diffusion-based method such as MAGIC, diffusion occurs when the transition Markov matrix (***M***) is exponentiated, as shown in Figure 1B. The exponentiated transition Markov matrix (***M*^t^**) step is a low-pass filter that increases the weighted affinities for similar data, whereas spurious neighbors are down-weighted [11]. The ***M*^t^** is a weighted graph that shows the probabilities of transition among samples, i.e., a single-cell sample in the case of scRNA-seq data, using random walks of any (*t*) length, rendering a temporal ordering of samples [42]. The information provided by ***M*^t^** has been used to impute data via diffusion on PCA space (as performed in MAGIC). In this case, data imputation is carried out by sharing information through local neighbors that follow data continuum densities [5]. However, there is evidence that preprocessing scRNA-seq data with MAGIC distorts the low-dimensional manifold by grouping many distinct cell types [40]. Thus, it is imperative to evaluate the distortion effect of diffusion on different manifolds embedded in the ***M*^t^** computed from MAGIC and sc-PHENIX. To accomplish that, we analyzed the visualization of the ***M*^t^** using multidimensional scaling (MDS) projections, which were described in more detail in the methods. Here, we refer to the MDS plot of the ***M*^t^** computed from PCA initialization as **_PCA_*M*^t^** and PCA-UMAP initialization as **_PCA-UMAP_*M*^t^**.

To visually compare the information embedded in the ***M*^t^** between the two methods, we used two different datasets: (1) the Modified National Institute of Standards and Technology (MNIST) database of handwritten digits [43] (Figure 4); this dataset used it as quality control because it has a cluster structure embedded in a low-dimensional space [43]. Moreover, the composition of clusters and trajectories can be visually evaluated as images on the low-dimensional manifold. (2) The neuronal dataset from the adult mouse visual cortex (Figure 5) from Tasic et al. [39] contains several (21) cell phenotypes characterized through experimental procedures (FACS-isolation) and in silico [39]. We used the neuronal dataset to quantify the preservation of the local structure (also known as cluster structure) through different combinations of parameters with MAGIC and sc-PHENIX. Additionally, the dataset from Tasic et al. [39] has already been used in other work to establish that MAGIC’s imputation groups several phenotypes [40], and we are going to explain the reason for this distortion with MAGIC and how to mitigate it with sc-PHENIX in terms of local and global data structure preservation.

For this section, an accurate diffusion process for these datasets will preserve a good local, global, and continuum structure in the ***M*^t^** MDS plots. If the data structure is poorly maintained in the ***M*^t^** MDS plots, it will result in overlapping and/or wrong disposition of distinct clusters and samples. Thus, there is a loss of data structure. For example, in the MDS plots of the PCA space in both datasets, we observed that the densities of the different digit number images (Figure 4A and Appendix A) and cell phenotypes (Figure 5A) are close to each other in the PCA space. Furthermore, some clusters are only partially separated by minor breaches with distinct sample overlapping (as noisy visualizations), for example, in Appendix A of the MNIST dataset. This is due to the undesired effects of distance concentration, where paired distances among samples in high-dimensional space tend to become more similar [8,9]. On the other hand, **_PCA_*M*^t^** revealed a better fine-grained local structure in the MNIST data; for example, see 2D **_PCA_*M*^t^** (Figure 4D) and 3D **_PCA_*M*^t^** (Figure 4B and Appendix A). 

Additionally, for the scRNA-seq data, the PCA ***M*^t^** (Figure 5C) was better in local structure preservation than the MDS plots of the PCA space (Figure 5A). However, in those figures, the local structure gets lost due to the connection between neighboring regions of different cluster densities. Consequently, the local and continuum structures are lost. For example, we can see in Figure 4F and Appendix A that the images of the number “6” change their shape gradually through the data continuum in a single branch (red line in Figure 4F). However, the “6” digit cluster density is connected with all digit cluster densities in the center of the manifold. 

In contrast, in **_PCA-UMAP_*M*^t^**, the “6” digit cluster is separated from distinct digit clusters; see for 2D **_PCA_*M*^t^** (Figure 4E and Appendix A) and 3D **_PCA-UMAP_*M*^t^** (Figure 4C and Appendix A). We plotted the handwritten images on the 2D **_PCA-UMAP_*M*^t^** manifold; we observed locally that the cluster is integrated by branches that show the inner differences of shape changes of the “6” digit images (three red lines in Figure 4G) and all digits (Appendix A). Based on this evidence, **_PCA-UMAP_*M*^t^** better represents the local and the continuum structure of the MNIST dataset. 

A similar conclusion regarding the local structure is obtained for **_PCA_*M*^t^** (Figure 5C) and **_PCA-UMAP_*M*^t^** (Figure 5D) for the neuronal scRNA-seq dataset. For **_PCA_*M*^t^** (Figure 5C), the local structure is not well preserved because peripheral regions of different cluster densities are connected, making it challenging to identify information about the heterogeneity of several cell phenotypes. However, local structures are better preserved in **_PCA-UMAP_*M*^t^** (Figure 5D) than in **_PCA_*M*^t^** (Figure 5C).

Concerning the local structure loss in the 2D **_PCA_*M*^t^** of MNIST (Figure 4E and Appendix A) and 3D **_PCA_*M*^t^** (Figure 4C and Appendix A), one hypothesis could be that handwritten numbers with similar shapes are located in nearby regions between the densities of distinct digit clusters. However, there are no similar shapes among digit images at the center of the manifold (Appendix A), thus proving the loss of the local structure.

Regarding local structure preservation in the **_PCA_*M*^t^** of scRNA-seq data (Figure 5C), there is a distortion of the manifold due to local structure loss; it is because GABA and glutamatergic neuronal cells are grouped in the manifold. It is well known that GABAergic and glutamatergic neuronal cell phenotypes carry out divergent neuronal functions [39]. Thus, these clusters could not be nearest neighbors in the manifold shown in Figure 5C. Moreover, some non-neuronal phenotypes are nearest neighbors of the neuronal cells (greenish colors in Figure 5C). Mukherjee et al. showed similar distortion in t-SNE projections from the imputed data with MAGIC [40] for this neuronal dataset.

Regarding the global structure present in the MNIST data, we concluded that the PCA MDS plot, **_PCA_*M*^t^** and **_PCA-UMAP_*M*^t^** render similar results to preserve the global structure, see Appendix A, Figure 4D,E, respectively. The disposition of clusters from these methods can be used as a reference to determine the global structure [11]. For instance, digits 4, 7, and 9 have similar shapes, implying a challenge to project the digits in separated clusters [44]. Moreover, given the similarity, the disposition of these clusters should be close to each other to preserve the global structure. This is expected because the MDS, PCA, and the combination of both are global approaches [11,45]. 

Concerning the global structure in the scRNA-seq dataset, the MDS plot of the PCA space (Figure 5A) shows that the non-neuronal and neuronal phenotypes are ordered well from top to bottom, giving a general overview (non-neuronal cells, glutamatergic, and then GABAergic). Due to PCA and MDS being global approaches, it is not surprising that the local structure is lost. Thus, all neuronal subtypes are not well separated in the MDS plot of the PCA space (Figure 5A). Concerning the global structure of the **_PCA_*M*^t^**, it is not well preserved; this effect is observed in the center of the **_PCA_*M*^t^** manifold (Figure 5C). The global structure loss is due to the disposition of several non-neuronal clusters (in greenish) in the neuronal cluster space (blue and red-yellowish) between the GABAergic and glutamatergic neuronal cell phenotypes overlap. In contrast, in **_PCA-UMAP_*M*^t^** (Figure 5D), the disposition of non-neuronal cells is far away from the neuronal phenotypes, and the subclusters are more defined and contained. Only the UMAP plots (Figure 5B) and our approach (Figure 5D) generate better dispositions of the three main clusters from the neuronal dataset without compromising the local structure. However, when **_PCA-UMAP_*M*^t^** compared to UMAP plots in the **_PCA-UMAP_*M*^t^** manifold, non-neuronal clusters are more defined and separated from other neuronal phenotypes in the periphery of the manifold, preserving a better global structure.

Finally, we visually evaluated the continuum structure from the manifold of ***M*^t^** plots using the MNIST dataset and observed the shape continuum along densities. We did not evaluate the continuum structure in the neuronal dataset with an MDS visualization because the dataset has several unconnected cell phenotypes with few samples in each one. This is a condition that is not adequate for analyzing the continuum of gene expression dynamics, but it is suitable for studying local and global structures. Additionally, it requires pseudotime methods. This is problematic because the implementation of several pseudotime methods requires different strategies for presenting gene expression dynamics along trajectories [46], which is especially difficult if we want to evaluate (*t*, *knn*, and PCA dimensions) parameter combinations of MAGIC and sc-PHENIX, as we evaluated in a further subsection but only taking into account the local and global structures. 

#### 3.3.2. Evaluation of Continuum Structure Preservation after MAGIC and sc-PHENIX on Gene–Gene Interaction Visualizations

In terms of scRNA-seq data imputation, the preservation of the continuum structure has been visualized and analyzed from imputed gene–gene interactions in the work of van Dijk et al. [5]. Therefore, we visually evaluated the data structure from the imputed gene–gene interactions from the 10x human peripheral blood mononuclear cells (PBMCs) scRNA-seq dataset [47] to evaluate the continuum structure, see Appendix A. In the PBMC dataset, there are nine characterized immune phenotypes (and contain sufficient cells to analyze the transition of a naive CD4 + T to a memory CD4 + T state (its corresponding markers are CCR7 and ILR7). This transition implies well-known CCR7 downregulation (references in Appendix A). In brief, we observed in the CCR7-ILR7 gene interactions that the CCR7 downregulation was distorted by MAGIC due to over-smoothing CCR7 and ILR7 expression to other distinct phenotypes, compromising the continuum structure. In contrast, with sc-PHENIX using PCA-UMAP initialization, CCR7 downregulation was preserved in most of the parameter combination values. Thus, based on the recovered gene–gene interactions, we concluded that the imputation with sc-PHENIX captures the continuum structure of scRNA-seq data better than MAGIC based on the transition dynamics. Please refer to Appendix A for complete details, discussion, and analysis.

#### 3.3.3. Effect of Increasing Parameters and PCA Dimensions in the Cell Neighborhood Captured in the Exponentiated Markov Matrix Using MDS Visualizations

For Figure 4 and Figure 5, we chose only one combination of parameters for MAGIC and sc-PHENIX. At least with MAGIC, we chose the parameters that Dijk van et al. ensured the robustness of MAGIC’s imputation for different *knn* and *t* values and PCA dimensions [5]. In brief, they evaluated a pairwise comparison of R2 correlations of imputed data with different combinations of parameters. However, implications of data structure preservation are dismissed in that correlation analysis. In this subsection, we evaluate cluster structure (also known as local structure) from several 2D MDS plots of the ***M^t^*** based on distinct parameter configurations for MAGIC and sc-PHENIX. We continued using the neuronal dataset to demonstrate the robustness of cluster structure preservation under different parameter combinations, quantified by the accuracy of the KNN classifier (accuracy is the ratio of the number of correct predictions to the total number of predictions made). In Appendix A, we visually analyzed how increasing diffusion parameters such as *knn*, *t*, and the number of PCA dimensions affected the cell neighborhood on the ***M^t^***, along with its accuracy value. In Appendix A, we observed in several **_PCA_*M*** (MAGIC) plots that the three main clusters—GABAergic (red-yellowish), glutamatergic (blueish), and non-neuronal (greenish) cell types—overlap, losing local structure with various combinations of diffusion parameters. Additionally, we noted that local structure is lost as one increases the values of the *knn*, *t*, and PCA dimensions parameters. This results in unrelated cell phenotypes overlapping or being near each other, making it difficult to separate the distinct cell phenotypes (loss of the local structure). This indicates that the manifold is heavily distorted due to diffusion on the PCA space with increasing PCA dimensionality, *knn*, and *t*. Mukherjee et al. [36] obtained a similar conclusion about the distortion with MAGIC for the same scRNA-seq dataset and others. In contrast, in various **_PCA-UMAP_*M*^t^** plots (sc-PHENIX), the detrimental effects are diminished, capturing the local structure; we observed this by a clear separation of distinct clusters, especially for GABAergic and glutamatergic clusters. 

In this study, we evaluated the performance of MAGIC and sc-PHENIX using a *knn* classifier to determine the accuracy of cluster structure preservation under different parameter configurations, including the number of PCA dimensions, *t*, and *knn* values. To draw general conclusions, we visualized the accuracy metrics for a total of 250 combinations of 2D MDS plots, with 125 from MAGIC and 125 from sc-PHENIX. The results presented in Figure 6 reveal several key insights.

Firstly, sc-PHENIX consistently demonstrated higher accuracy compared to MAGIC across most parameter configurations. This is evidenced by the predominant presence of red and orange areas in the accuracy plots for sc-PHENIX, indicating better performance. Notably, sc-PHENIX achieved its highest accuracies with higher PCA dimensions (100, 200, and 500) and moderate-to-high *knn* values, particularly as *t* increased. This suggests that sc-PHENIX is more effective at preserving local cluster structures under these conditions. The data embedding for each parameter set and method is presented in Appendix A.

In contrast, MAGIC’s performance was generally lower, as indicated by the green and blue areas in the accuracy plots. The accuracy of MAGIC decreased significantly with lower *knn* values and higher PCA dimensions, and increasing *t* did not result in consistent improvements; MAGIC’s default *t* value is 3. This pattern highlights the limitations of MAGIC in maintaining accurate cluster separations under varying parameter settings.

Overall, these findings suggest that sc-PHENIX is more robust than MAGIC in preserving the local structure and achieving higher accuracy across different parameter combinations. The best results for sc-PHENIX were observed with higher PCA dimensions and higher *knn* values, while MAGIC struggled to maintain high accuracy under similar conditions. These insights underscore the importance of parameter selection in optimizing imputation methods for single-cell RNA-seq data analysis.

#### 3.3.4. Distribution of Distinct Cluster Samples on Dense Regions of ***M*^t^**

To incorporate additional criteria beyond the MDS approach, we quantified the distribution of MNIST images on the exponentiated Markovian matrix (***M*^t^**) to directly detect dense regions (Figure 7). Thus, we can eliminate a possible bias of the MDS methodology towards sc-PHENIX, as suggested in Figure 7. Therefore, we could demonstrate that the spurious agglomeration of distinct clusters observed in multidimensional PCA (Figure 4A and Appendix A) and MAGIC’s ***M*^t^** (Figure 4D) was due to the detrimental effect of distance concentration and/or distortion of the data structure. To generate Figure 7, we used HDBSCAN [17] on the distance matrix from PCA space, and each of the ***M*^t^** was constructed from PCA space and PCA-UMAP space. HDBSCAN determines noise to samples that do not belong to a dense region. 

This analysis aimed to identify dense regions that indicate an agglomeration of similar MNIST images. This is achieved by accurately approximating the underlying manifold embedded in the ***M*^t^** graph or from the multidimensional PCA space. If the dense regions show several dissimilar images, then the construction of the ***M*^t^** graph or multidimensional PCA space fails to capture the true nature of the underlying structure. To achieve this, we used HDBSCAN [17] on the distance matrix from PCA space, and each distance matrix generated from the ***M*^t^** is constructed from PCA space and PCA-UMAP space. HDBSCAN associates samples that do not belong to a dense region as “*noise*”. In our case, this provides one way to measure the quality of the density-based clustering in the embedding space, whether it is in the multidimensional PCA space or the ***M*^t^** generated from MAGIC or sc-PHENIX. Another way to qualify the quality of the clustering is in the context of natural clusters in MNIST data; samples often have similar shapes. For example, handwritten numbers 4 and 9 share similar shapes, and density-based clustering methods can group them into a single dense cluster. However, a gap or a low-density region may interrupt the density continuity between similar numbers. That seems to be the case for **_PCA-UMAP_*M*^t^** generated from sc-PHENIX; the number 7 appears to be separated from 4 and 9 only by a low-density region that has few samples and is apparently disconnected, as depicted in Figure 4E and Appendix A. Nevertheless, samples of digits 4, 7, and 9 are still relatively close to each other compared to the rest of the digits, preserving a well global structure.

Regarding the results obtained by HDBSCAN, in PCA space (Figure 5A), we detected only one dense cluster integrated by the number “1” of the MNIST samples, while most of the other MNIST samples are considered noise (Figure 7A,D). This result converges with the MDS plot (Figure 4A), samples of the digit “1” that are more visually dense than the rest of the numbers. This can be observed in the interactive plot shown in Appendix A: Appendix A.

In diffusion on PCA space, one dense cluster is detected (Figure 7B,E, Cluster A). This cluster has almost all MNIST samples. However, a few MNIST samples are considered noise (Figure 7B,E) compared to the PCA manifold (Figure 7A). In the case of diffusion on PCA-UMAP space, the distribution of densities of distinct MNIST images are better separated into seven clusters (A–G clusters), as shown in Figure 7C,F. This non-visual criterion (HDBSCAN on ***M*^t^** graph) indicates that the ***M*^t^** computed from sc-PHENIX with PCA-UMAP initialization can separate true distinct dense clusters (MNIST digit numbers).

Our alternative non-visual analysis (Figure 7) depicts information about the multiple MDS embeddings of MAGIC’s ***M*^t^** (Figure 4), revealing that several distinct phenotypes were erroneously grouped due to the inadequate PCA initialization for the construction of the ***M*^t^** rather than through the use of the MDS methodology.

#### 3.3.5. Comparing the Visual Results of PHENIX (PCA-UMAP) Versus Specialized Visualization Methods

Additionally, we wanted to see the difference in the preservation of local, global, and continuum structure of the data between our visualization method (MDS plots of the ***M*^t^** from diffusion on PCA-UMAP space of sc-PHENIX) and Potential of Heat Diffusion for Affinity-based Transition Embedding (PHATE). PHATE plots of the neuronal dataset (Appendix A) and non-neuronal cells (green cluster) are overlapped with neuronal phenotype clusters. Thus, the local structure is lost. Additionally, PHATE applied to MNIST data (Appendix A) preserved data structures better than MAGIC (Figure 4D and Appendix A) but not better than sc-PHENIX (Figure 4E and Appendix A). Although PHATE separates MNIST and neuronal clusters better than MAGIC, the local structure becomes compromised. Consequently, samples of various handwritten numbers are grouped together in the PHATE manifold, thus affecting the continuity of the data. The effect can be seen at the extreme of the dense regions near distinct numbers; the continuity of the gradual transition of shape numbers is interrupted by different numbers that do not share either the shape (Appendix A) or its digit label (Appendix A). For example, in Appendix A, at the extreme periphery of the dense region of 1-digit clusters, 1 s are connected to 7 s and 2 s. However, there is no gradual shape transition between 7 s and 2 s. Additionally, 7 s and 2 s near the periphery of the dense region of 1-digit samples do not have similar shapes, even though PHATE was designed to preserve local and continuum structures. Using PHATE, the global structure of the MNIST dataset is preserved, as 7 s, 4 s, and 9 s are close to each other, and the same is true for 3 s, 5 s, and 8 s; these are global features of the MNIST dataset [44]. However, this is not true for the neuronal dataset, as non-neuronal phenotypes are grouped together with the two main neuronal phenotypes in the 2D PHATE plot (Appendix A).

Additionally, in UMAP (Appendix A), clusters of digits 4, 7, and 9 are merged into a single large cluster, which makes it difficult to observe any substructures within each digit cluster (Appendix A). It is well-known that UMAP is unsuited for preserving continuum structure data [11], especially in 2D dimensions [27]. In contrast, sc-PHENIX can preserve the continuum structure better than UMAP and, at the same time, preserve the local structure, as evidenced by the distinct inner branches for each digit in sc-PHENIX (Appendix A).

#### 3.3.6. The inherent Risk of Diffusion Artifacts

Previous sections show that the connection between densities of distinct clusters (images or cell phenotypes) is an artifact when the diffusion is accomplished on a poorly suited manifold initialization (such as PCA space). This artifact is driven by a small gap between the densities of distinct clusters in PCA space, reflected in the poor separation of distinct clusters (Figure 4D and Figure 5A and Appendix A). Here, we showed that PCA does not separate densities from distinct clusters well enough to avoid spurious connections by diffusion (Figure 4A). The most probable reason is that sparsity significantly impacts the nearness of densities from distinct clusters in PCA space; we prove this in Appendix A. Additionally, diffusion methods approaches based on PCA space (PHATE) distort the manifold when data have high rates of missing values, whereas PCA-UMAP appears to be more suitable for separating distinct clusters for sparse data (Appendix A). According to our results, PCA-UMAP initialization solves this major issue for accurately constructing ***M*^t^** using only PCA initialization. This is important because sc-PHENIX and MAGIC require the manifold embedded in the ***M*^t^** for its final imputation step (Figure 1B, matrix multiplication). Therefore, to avoid over-smoothing (an effect caused by sharing information among spurious nearest neighbors), the ***M*^t^** needs to capture the data’s local, global, and continuum structure as much as possible. For that, we recommend sc-PHENIX, using PCA-UMAP (Figure 4D) space instead of only PCA to solve the major issues caused by MAGIC using diffusion on PCA space (Figure 4C).

The importance of preserving data structures in diffusion methods, or any other manifold learning approach applied to scRNA-seq datasets, is the detection and disposition of rare phenotype cells among ubiquitous phenotype cells in the manifold space, also known as transcriptomic space. Rare phenotypes are in low-density regions of cells on a manifold space because of a large acceleration of gene expression of cells due to a phenotype transition; these rare populations are present for a short period of time [48].

### 3.4. Evaluation of Over-Smoothing among Distinct Cell Phenotypes 

To evaluate over-smoothing, we continued using the adult mouse visual cortex cell dataset from the previous section because we knew beforehand the ground truth about the differential marker among phenotypes [39]. The dataset’s main cell phenotypes are GABAergic, glutamatergic, and non-neuronal, with an additional 21 sub-phenotypes belonging to these main cell types. We expect that a good imputation will not over-smooth these markers to distinct cell phenotypes. Hence, we evaluated the over-smoothing effect of specific cell phenotype gene markers using MAGIC and sc-PHENIX. The gene markers used were vascular endothelial growth factor receptor 1 (*Flt1*), vasoactive intestinal peptide (*Vip*), choline acetyltransferase (*Chat*), somatostatin (*Sst*), and serpin family B member 11 (*Serpinb11*). For instance, the *Flt1* marker corresponds to endothelial and smooth muscle cells, while *Vip* and *Chat* markers are for Vip cells, *Sst* marker for Sst cells, and *Serpinb11* marker for Gluta_L6b [39]. It is important to mention that other imputation benchmarking evaluated over-smoothing using phenotype markers as the gold standard but only using default parameters, thus making the evaluation biased. Therefore, we used several combinations of parameters (*t*, *knn*, and PCA dimensions) to evaluate MAGIC and sc-PHENIX performance.

In Figure 8A–C, we observed the recovered *Flt1* marker expression using MAGIC and sc-PHENIX, as well as the non-imputed expression on the UMAP plot from the original data. In Figure 8, MAGIC and sc-PHENIX have different outcomes regarding over-smoothing along distinct cell phenotypes using different increasing combinations of PCA dimensions, *knn*, and *t*. We observed in Figure 8A that increasing *knn*, *t*, and PCA dimensions, with MAGIC, increases over-smoothing compared to sc-PHENIX. This is because MAGIC does not maintain the expression of gene markers in their respective cell phenotype clusters visualized in the UMAP projection. However, sc-PHENIX maintains the expression of gene markers shown in Figure 8A. We obtained similar results for the *Vip*, *Sst*, and *Chat*, as shown in Appendix A.

In addition, we quantified the imputation performance of MAGIC and sc-PHENIX to maintain the differential expression of specific gene markers in their respective cell phenotypes (Figure 9). At the same time, the evaluation considered the effect of several combinations of diffusion parameters (*knn* and *t*) and PCA dimensions. Thus, we evaluated the precision, recall, and f1-score metrics. We used the *Flt1* marker for NonNeu_Endo cells and NonNeu_SMC cells, *Sst* marker for GABA_Sst cells, *Chat* marker for GABA_Vip cells, *Serpin11* for Gluta_L6B cells type. We observed that the imputation performance decreased using MAGIC as we increased the parameters and PCA dimensions. Under the same evaluation, sc-PHENIX does not jeopardize the performance (Figure 7) by avoiding over-smoothing, as observed in the ***M*^t^** MDS plots. Consequently, sc-PHENIX correctly separates densities of distinct cell phenotypes (GABAergic, glutamatergic, and non-neuronal cells) by PCA-UMAP initialization (Figure 5D). In contrast, MAGIC with the PCA initialization does not achieve a good imputation performance with any combination of parameters; all markers are over-smoothed, even with local imputations (low values of *knn* and *t*), as shown in Figure 8 and Figure 9. The same artifacts have been reported in other evaluations using MAGIC, which groups together different cell types [40].

### 3.5. Evaluation of the Heterogeneity of Spheroids of MCF7 Cells Data with Imputation

To evaluate the recovered biological insights post-imputation in a comparative analysis with MAGIC and sc-PHENIX, we imputed the 3D Multicellular Cancer Tumor Spheroids of breast cancer cell line MCF7 dataset (MCF7 MCTS) from [49]. These sc-RNAseq data comprise the transcriptome at two instants of MCF7 MCTS growth (6 and 19 days). The authors identified three subpopulations with a non-imputation approach and characterized their functions through a Gene Set Enrichment Analysis (GSEA). In brief, these subpopulations were classified as invasive, proliferative, and reservoir (an intermediate phenotype of the previous). The invasive phenotype had immune-adaptive characteristics, the proliferative phenotype promoted cell growth, and the reservoir subpopulation might be a putative transition phenotype, sharing proliferation and invasive pathways.

#### 3.5.1. Differences in 3D PCA Manifold of Imputed Data

In this subsection, to identify and characterize the extreme phenotypic states and the transition phenotypes in a better-formed structure [5], see Section 2.7.4 for the definition of extreme phenotypic states. We compared the 3D-PCA plots of the imputed data with MAGIC and sc-PHENIX in Figure 10A and Figure 10B, respectively. For more details from the previous plots, see its interactive 3D-PCA plots for MAGIC: Appendix A and sc-PHENIX: Appendix A. We detected and compared the dense clusters obtained by HDBSCAN in each 3D-PCA space from both methodologies. We observed five dense clusters in the PCA space with MAGIC (Figure 10A). Meanwhile, with sc-PHENIX (Figure 10B), there are ten dense clusters. 

Dense clusters with sc-PHENIX (Figure 10B) are visually more separated than MAGIC (Figure 10A). Therefore, fewer dense clusters were detected in the PCA space post-MAGIC imputation (Figure 10A) compared to sc-PHENIX (Figure 10B) due to MAGIC’s tendency to group cells together. Consequently, we interpreted this nearness of the clusters as likely similar phenotypes in the 3D PCA manifold in Figure 10A and Appendix A.

In contrast, the far distances on the 3D-PCA plots post-sc-PHENIX (Figure 10B) can be interpreted as distinct cell phenotypes. Additionally, with sc-PHENIX, we observe more branches in Figure 10B and Appendix A, all cells, including the non-dense cells (noise samples with HDBSCAN), compared to MAGIC (Figure 10A and Appendix A). While there is not a clear trajectory structure with MAGIC, there is only a simple trajectory, especially when viewed with PC1 and PC2. Four branches are well defined in sc-PHENIX, which could depict the intricate cell transition states among distinct phenotypic states of the MCF7 MCTS. 

Regarding the extreme phenotypes with sc-PHENIX, at first glance, we can see that the 3D PCA plot has four extremes corresponding to clusters 2, 4, 5, and 7. In contrast to MAGIC, clusters are grouped primarily in the small range along the PC3 axis. However, when examining PC1 and PC2, clusters 0, 2, and 4 could be considered as extreme phenotypes (Appendix A). 

In the following subsections, we biologically characterized all dense clusters of MCF7 MCTS to confirm the biology involved, especially focusing on the extreme phenotypes and its possible intermediate states.

#### 3.5.2. Over-Smoothing Analysis of Imputed Data and Continuum Structure Implications

As mentioned in the previous subsection, the clusters shown in Figure 10A (MAGIC) are closer than those projected in Figure 10B (sc-PHENIX), which could indicate that the gene expression profiles between clusters are similar, as shown in Figure 10B. We corroborated this by using the EMD score to detect differentially expressed genes by clusters. After applying the EMD score, we identified the shifts in the gene expression among dense clusters in MAGIC (Figure 10A) and sc-PHENIX (Figure 10B) in terms of DEG. In agreement with the visual closeness observed in Figure 10A and Appendix A (MAGIC) with the 3D-PCA plots, the EMD score (Figure 10A DEG heatmap) reproduces the closeness of their gene expression profiles among clusters 0–3. In contrast, sc-PHENIX helps to identify more separated clusters in the 3D-PCA plots (Figure 10B and Appendix A), thus indicating a higher variability in their gene expression profiles (Figure 10B DEG heatmap).

Additionally, with MAGIC in Figure 10A (DEG heatmap), mitochondrial genes (MT genes) are over-expressed in clusters 0–3 (4 of 5 dense clusters). However, with sc-PHENIX shown in Figure 10B (DEG heatmap), only the 0–2 clusters (3 of 10 dense clusters) of the MT genes are over-expressed. To confirm that the over-smoothing with MAGIC was not caused by a large sample size from HDBSCAN clusters that masked the heterogeneity of cell phenotypes (as shown in Figure 10A DEG heatmap), we reduced the sample size per cluster by creating eleven k-means clusters. This revealed the same over-smoothing pattern as that shown in Figure 10A; more details are shown in Appendix A. Thus, with MAGIC, the MT genes (mitochondrial genes) and several DEG are over-smoothed among the data shown in Figure 10A (DEG heatmap), while with sc-PHENIX, this was not the case, as shown in Figure 10B (DEG heatmap). 

Furthermore, we observed with sc-PHENIX that vascular endothelial growth factor A (*VEGFA*) over-expression is present mainly in cluster 2 (Appendix A and MT genes (Figure 10B, of DEG heatmap). With MAGIC (Appendix A), cluster 4 over-expressed *VEGFA*. However, cluster 4 has non-differentially expressed MT genes (Figure 8A, DEG heatmap), even when the MT genes are over-smoothed among the most dense clusters. The identification of MT gene expression is important in this case because it can help to identify necrotic cell death phenotypes in scRNA-seq data. Necrosis is morphologically defined by cell and organelle swelling, early plasma membrane rupture, and the spilling of cellular material [50]. Necrosis is generally considered to be a passive process because it does not require new protein synthesis and has only minimal energy requirements [51]. Additionally, it is well known from scRNA-seq procedures that cells in tumor samples with increased mitochondrial gene expression are necrotic [52]. Thus, cells with high mitochondrial transcripts and low Unique Molecular Identifier (UMI) counts most likely represent a dying cell population. Regarding *VEGFA* expression, this gene is substantially enhanced in areas surrounding necrotic foci in tumor spheroids, suggesting a mechanism by which a hypoxic microenvironment might stimulate genes involved in tumor angiogenesis [53,54]. Only a post-imputation process with sc-PHENIX could detect the dying cell populations with lower UMI counts (Appendix A), the over-representation of MT genes, and *VEGFA* expression. All these results are indicators of a heavily injured and necrotic state in clusters 0, 1, 2, and 3, where cluster 3 is the most representative for necrosis due to the over-expression of MT genes and *VEGFA*, with a lower UMI count.

Now, diffusion methods, such as MAGIC and sc-PHENIX, allow us to reveal the state of data continuity [5]. We used gene–gene interactions and 3D PCA plots from the imputed MCF7 MCTS dataset to test our hypotheses and identify cell-state continuums among extreme clusters. We detected the transition states among extreme clusters for imputation methods, but the results differed among methods, as shown in Figure 10 and Figure 11, Appendix A, and Appendix A. For example, based on the 3D PCA plots with sc-PHENIX, we visually identified four branches or trajectories to the extreme clusters; we do not consider directionality: (1) 4 to 2, (2) 4 to 7, (3) 7 to 5, and (4) 3 to 2, see Appendix A, and Figure 10B and Figure 11. In contrast with MAGIC, we detected only one branch that follows the sequence from clusters 0 to 1 to 2 to 3 to 4; see Appendix A and Figure 10A. We detected one branch with MAGIC because of the over-smoothing of data (Figure 10 DEG heatmap); DEG is similar among clusters. Here, we do not evaluate directionality; we acknowledge that RNA velocity methods based on imputation are separate projects.

Now, based on the *VIM-CDH-FN1* interaction with sc-PHENIX, extreme cluster 4 indicates that there could be a mesenchymal population where there is down-regulation of *CDH1* (*e-cadherin*) and upregulation of *VIM*, as shown in Appendix A. Additionally, in the next subsection, cluster 4 was characterized as a proliferative phenotype (Figure 9 and Figure 10B). We identified that the down-regulation of *CDH1* is correlated with cells that over-expression of *ANLN* (Anillin) (Appendix A) and *MKI79* (Marker of Proliferation Ki-67) (Appendix A) markers, as a proliferative state that show a mesenchymal state.

In contrast, the *VIM-CDH1* interaction recovered by MAGIC (Appendix A) shows that *VIM* and *CDH1* are proportionally increased. This contradicts the literature [55] because *VIM* is downregulated when *CDH1* is upregulated [56]. Again, this is a wrong relationship due to the detrimental effect of over-smoothing. Thus, all indicate that sc-PHENIX captures the gene expression dynamic involved in a proliferative state more accurately than MAGIC.

Additionally, in Appendix A with MAGIC, we visualized all evaluated genes *ANLN* (involved in proliferation [49]), *MT-ND3* (mitochondrial gene, necrotic state), and *S100A8* (involved in a hypoxic/invasive state [49]) genes on the *VIM-CDH1-FN1* interaction and the 3D-PCA plots, the genes are over-smoothed having a strong weight in several cells. However, with sc-PHENIX, over-smoothing does not have a strong weight, and gene expression seems to be confined in the nearness of the local cell neighborhood (Appendix A).

With sc-PHENIX, we observed a more diverse differential gene expression and transition states among extremely dense clusters than in the case of MAGIC. This constitutes an intuitive criterion to support the functional cluster in the embedding space of sc-RNAseq data. As we have exposed in this section, sc-PHENIX has a higher resolution than MAGIC in terms of detecting diverse phenotypes with a reduced rate of over-smoothing. In the next subsection, we will discuss how these improvements allow us to obtain rich biological insights that explain the environment of MCF7 MCTS.

#### 3.5.3. GSEA Analysis of Cell Phenotype

Previous results showed that the imputation of MCF7 MCTS data with sc-PHENIX uncovered several genes that might play a role in depicting the functionality of each subpopulation within the tumor model. To associate and gain information about the triggered biological pathways, we applied GSEA (Gene-Set-Enrichment Analysis) to the EMD score value of each dense cluster for MAGIC and sc-PHENIX, see Figure 10A,B GSEA heatmaps, respectively. The GSEA associates the under and over-expressed genes with a curated database. With non-imputation, the previous work obtained 21 HALLMARK pathways. We obtained 41 HALLMARKS pathways with MAGIC, and with sc-PHENIX, we obtained 45 HALLMARKS. Therefore, with imputation strategies (Appendix A), we obtained more than 20 new HALLMARKS pathways with MAGIC and sc-PHENIX. It is important to mention that over-smoothing with MAGIC does not lead to more HALLMARKS pathways compared to sc-PHENIX. Thus, indicating that sc-PHENIX shares gene information among true redundant (similar) cells leads to the unmasking of more HALLMARKS pathways without unnecessary over-smoothed data. sc-PHENIX revealed four new HALLMARKS pathways: XENOBIOTIC_METABOLISM (cluster 6 and 10), WNT_BETA_CATENIN_SIGNALING (cluster 5), APICAL_JUNCITON (cluster 3 and 6) AND COAGULATION (Cluster 1 and 3), which are all depicted in Figure 10B and Appendix A. These HALLMARKS do not appear in either the original work (non-imputation) or MAGIC.

Based on the HALLMARKS of the original work [49] and this work, we determined a few HALLMARKS for (1) invasive phenotype HALLMARKS related to invasive and pro-survival roles: HALLMARK of REACTIVE_OXYGEN_SPECIES_PATHWAY, INTERFERON_ALPHA_RESPONSE, INTERFERON_GAMMA_RESPONSE, INFLAMMATORY_RESPONSE, APOPTOSIS, HIPOXIA, among others, (2) the proliferative phenotype had HALLMARKS, such as E2F_TARGETS, G2M_CHECKPOINT, MITOTIC_SPINDLE, GLYCOLYSIS and OXIDATIVE_PHOSPHORYLATION, among others, (3) transition phenotype state, with proliferative and invasive HALLMARKS, and (4) a necrotic state, with few HALLMARKS pathways compared to the previous phenotypes.

With MAGIC, the proliferative phenotype corresponds to clusters 0–3, and the invasive phenotype is cluster 4 (Figure 10A). Probably, due to the over-smoothing with MAGIC, there is no clear transition of the invasive and proliferative phenotypes in the dense clusters. For the above, we had to recall from the previous subsection that clusters 0–3 of the MT gene are over-expressed (Figure 10A DEG heatmap), and we observed that the proliferation HALLMARKS are enriched in clusters 0–3 (Figure 10A). Additionally, with MAGIC, dense clusters with few HALLMARKS were not found due to the necrotic state. This does not explain the cells in a microenvironment state and the context of factors, such as hypoxia and nutrient deprivation, at least with MAGIC. The side effects of over-smoothing can mask biological processes. The intratumoral heterogeneity results from a combination of extrinsic factors from the tumor microenvironment and intrinsic parameters from cancer cells themselves, including their genetic, epigenetic, and transcriptomic traits, their ability to proliferate, migrate, and invade, and their stemness and plasticity attributes [57]. It is important to ensure that imputation does not over-smooth the data, as this could obscure the heterogeneity involved in phenotypes within a microenvironment.

On the other hand, with sc-PHENIX based on GSEA analysis, we have a larger repertoire of extreme phenotypes that are involved in MCF7 MCTS; this is depicted in Figure 10B and Figure 11. For example, in Figure 11 we observed that the decrease in HALLMARKS pathways correlates with clusters 0, 1,2, and 3 clusters, in which MT genes and *VEGFA* (Appendix A) are over-smoothed, all indicating that these clusters could be necrotic states. Cluster 3 is the least represented for this necrotic phenotype, which seems to be a transition from cluster 10 and extreme cluster 4, presenting several inflammatory-related HALLMARK pathways (Figure 10B and Figure 11). Interestingly, both clusters 3 and 5 present APOPTOSIS HALLMARK pathways.

Based on the sc-PHENIX analysis shown in Figure 11, cluster 3 shares several enriched pathways with the invasive phenotype (cluster 5), which is an extreme archetype. Additionally, cluster 3 has the highest HIPOXIA NES value (3.7935) among all clusters (as shown in the Appendix A) and appears to be a transition state to cluster 2 (the transition from invasive to necrotic state, as seen in Figure 8B, i.e., transition of 3 → 1 → 2 clusters). This suggests that sc-PHENIX can capture phenotypes with survival mechanisms involved in wound healing that transition to a necrotic state. Furthermore, the COAGULATION HALLMARK (only detected with sc-PHENIX) was found to be enriched in cluster 3, and a large number of studies have confirmed that coagulation is positively correlated with angiogenesis-related factors in metastatic breast cancer [58]. Therefore, the results for cluster 3 indicate the presence of phenotypes involved in mechanisms of invasion related to the representation of angiogenesis genes. 

Regarding transition states with sc-PHENIX, extreme clusters 7 and 8 share some proliferative and invasive HALLMARKS. For example, just to mention a few of them, for the proliferation phenotype are the OXIDATIVE_PHOSPHORYLATION, ESTROGEN_RESPONSE_LATE, and MTORC1_SIGNALING pathways. For the invasive phenotype, there are TNFA_SIGNALING_VIA_NFKB, REACTIVE_OXYGEN_SPECIES_PATHWAY but not HIPOXIA pathways. The only HALLMARK that is representative for clusters 7 and 8 compared to the extreme clusters 4 (proliferative) and 5 (invasive) is the CHOLESTEROL_HOMEOSTASIS.

It is probable that the HALLMARK of CHOLESTEROL_HOMEOSTASIS has a crucial role in the cholesterol biosynthesis pathway in the maintenance of spheroids; this has already been reported [59]. Additionally, by using the REACTOME pathways database for GSEA, we find other unique pathways involved in the transition clusters 7 and 8 that are not present in the extreme proliferative and invasive clusters using sc-PHENIX. We found that clusters 7 and 8 had the highest NES values of the REACTOME_CELLULAR_RESPONSE_TO_STARVATION and REACTOME_MITOPHAGY (Appendix A), and these pathways are not present in the proliferative and invasive state extreme clusters. Additionally, REACTOME_MITOPHAGY is only present in clusters 7 and 8. Thus, it seems that these clusters, in particular, participate in a tumor microenvironment context that could be in an interlayer where the gradient of dispositions of nutrients begins to diminish. Additionally, mitophagy reduces overall mitochondrial mass in response to certain stresses, such as hypoxia and nutrient starvation. This prevents the generation of reactive oxygen species and conserves valuable nutrients (such as oxygen) from being consumed inefficiently, thereby promoting cellular survival under the conditions of energetic stress. It is worth mentioning that these transition clusters present OXIDATIVE_PHOSPHORYLATION. More analysis and experimental approaches need to be performed to corroborate this hypothesis for this model, which is not the scope of this article. These were only a few examples of the capabilities of the analysis downstream after sc-PHENIX. All these finer biological insights were not obtained with MAGIC or in the original work (non-imputation). Future work will bring more insights for these data based on imputation with sc-PHENIX.

## 4. Discussion

The imputation of scRNA-seq data is critical for recovering information lost due to noise in experimental protocols. However, all imputation methods, including those currently available, are susceptible to over-smoothing, highlighting the need for new strategies that more effectively recover lost data and enhance the interpretation of subsequent analyses [6,20]. In response, we introduce sc-PHENIX, an algorithm that builds on UMAP and diffusion maps theory to significantly mitigate over-smoothing and preserve the local, global, and continuum structures of scRNA-seq data. Our approach modifies the MAGIC method by incorporating a PCA-UMAP initialization, which enhances the imputation process’s accuracy and facilitates more efficient handling of missing values in high-dimensional data. This enhancement potentially elevates downstream analyses’ quality and reliability, yielding more precise and insightful biological interpretations. We rigorously evaluated sc-PHENIX and compared it with MAGIC across three different biological systems (as illustrated in Figure 5, Figure 6 and Figure 11 and in the Appendix A), as well as in non-biological systems (as shown in Figure 4). Our evaluations demonstrate that sc-PHENIX preserves the global and local data structures (Figure 4 and Figure 5) and maintains a continuum organization (Figure 4G, Appendix A). We tested the imputation performance with metrics using a variety of parameter combinations (*knn*, *t*, and PCA dimensions), finding that sc-PHENIX provides a superior solution to the problem of over-smoothing compared to MAGIC, as shown in Figure 9, and for the preservation of cluster structure, shown in Figure 6.

Additionally, we visually assessed the *knn*-smoothing method using the neuronal dataset to check for potential over-smoothing in 2D UMAP plots. The results indicated that *knn*-smoothing did not excessively smooth gene markers, as detailed in Appendix A. However, when applied to the PMBC dataset, as also discussed in Appendix A, it produced noise in gene–gene interactions akin to local imputation with MAGIC (Appendix A). This outcome likely stems from using PCA and its aggregation-type imputation as the input space. The *knn*-smoothing method primarily captured the general trends of the data, with local details being lost, as elaborated in Appendix A. In contrast, when applied to the PMBC dataset, SAVER was less effective at preserving the continuum of expression compared to sc-PHENIX, MAGIC, or *knn*-smoothing (Appendix A).

We demonstrated that sc-PHENIX effectively preserves data structure during imputation, enhancing the recovery of biologically relevant insights for the evaluated dataset (Figure 11). This capability allowed us to uncover previously hidden heterogeneity and state transitions, which would have remained undetected with non-imputation or MAGIC approaches. The significance of preserving these data structures is evident in identifying rare phenotypes located in low-density regions of the transcriptomic space manifold [48]. For instance, as shown in Figure 11, for the MCF7 MCTS dataset, sc-PHENIX enabled the detection of novel phenotypes not identified in previous studies, such as the extreme archetypes exhibiting a starving signaling state (cluster 7) and a necrotic state (cluster 2). Additionally, with sc-PHENIX, we discovered distinct trajectories that interconnected the extreme phenotype states, indicating that it is a continuum system. Not only that, but 3D PCA is well suited for finding extreme clusters and intermediate phenotypes imputed data by sc-PHENIX. We compared it with other methods that are commonly used for visualizing high-dimensional data, such as t-SNE, UMAP, and PHATE (Appendix A), only 3D PCA with sc-PHENIX was adequate for the disposition of the extreme clusters along with its intricate trajectories; other methods, despite their branches being preserved using t-SNE, UMAP, and PHATE, the use of 3D PCA reveals the extreme phenotype states on the manifold edges. The smoothing process using sc-PHENIX made the cells from their corresponding phenotypes more similar in terms of expression. As a result, 3D PCA allowed us to capture the global, local, and continuous structure more effectively, revealing trajectories and extreme phenotypes with high variability without the need to optimize points for visualizing high-dimensional data, as required by t-SNE, PHATE, or UMAP. This suggests that 3D PCA is the most suitable method for sc-PHENIX imputed data in terms of the biological information obtained. In contrast, applying t-SNE, PHATE, and UMAP to the imputed data with MAGIC did not yield clear results, showing only a single branch and lacking well-defined extreme phenotypes, as evident in Appendix A.

This is important to discuss because, for the imputed results obtained from the sc-PHENIX of the tumor spheroid dataset, cancer is typically considered a set of discrete phenotypic states in most reports. However, some authors support the idea that tumor cells reside within a cell-state continuum rather than in discrete subtype clusters [5,60,61]. For example, the EMT in scRNA-seq experiments can be analyzed as a continuum, where single cells follow a continuum in a low-dimensional manifold. In this manifold, the gene expression dynamics of EMT canonical markers, such as *VIM* and *CDH1* (e-cadherin), show the epithelial or mesenchymal states. In which the upregulation of *VIM* and the downregulation of *CDH1* indicate a mesenchymal state, and the opposite indicates an epithelial state [55]. Regarding tumor spheroids, mesenchymal cell populations normally give rise to other cells that are organized as three-dimensional masses rather than sheets [62,63]. Additionally, experimentally, hypoxia increases extracellular components involved in cellular adhesion, i.e., fibronectin (*FN1*) in tumor spheroids [64]. Finally, due to extracellular adhesive components, densely packed cells interfere with the oxygen supply to the tumors. This results in a gradient of oxygen concentration along with the tumor spheroid, and therefore, the presence of hypoxia inside the spheroids [65]. We can make hypotheses based on the spheroid dataset imputed by sc-PHENIX with the previous information.

Based on the examples discussed in the paper, we concluded that PCA-UMAP initialization for the diffusion process enables the preservation of the data structures in scRNA-seq data. Across all of the datasets analyzed, we observed that MAGIC progressively distorts data with increases in *knn*, *t*, or PCA dimensions, as illustrated in Figure 6. The distortion can be attributed to the inadequate separation of distinct clusters in the multidimensional PCA space, which leads to erroneous associations following the diffusion process (Figure 7G). Consequently, our findings indicate that PCA-UMAP initialization significantly mitigates distortion.

We demonstrated that by computing ***M*^t^** using PCA initialization, there is not enough separation to distinguish cell phenotypes, so MAGIC groups them together, as shown in Figure 7G. Moreover, its imputation implications are limited to sharing information with only a few local neighbors, referred to as local imputation by setting low-value parameters, as shown in the Appendix A, which can still lead to over-smoothing of data (Figure 9). Thus, MAGIC is prone to over-smoothing data when higher values of parameters are used. Moreover, while over-smoothing can be partially mitigated by sharing information with a few local cells by using lower values of parameters (also known as local imputation), there is an incomplete denoising of data and a failure to recover relevant biological insights. 

In the case of local imputation, gene–gene interactions are not well-structured or do not provide sufficient information regarding the dynamics of gene expression markers in cell states, local granularity, and continuum transitions (Appendix A). Therefore, with MAGIC and sc-PHENIX, there is a need to increase the value parameters to better reveal gene–gene interactions, but only to a certain extent, to avoid over-imputation (especially with MAGIC). However, due to the well-separated distinct phenotypes and the preservation of continuum, sc-PHENIX can achieve a more complete network of cells, revealing hidden divergent branches derived from the principal branch. This is especially important for rare cells that experience transcriptional bursting, particularly over short periods of time. This activity could be reflected in manifolds as low-density cell states [66]. 

Furthermore, the negative effects of local imputation can be found in the gene–gene interaction recovered by MAGIC in the work of van Dijk et al. [5]; see the gene–gene interactions with starting few values of *t* steps. Dijk et al. showed that the hidden structure of gene–gene interactions became apparent as the *t* value increased. However, they did not analyze and discuss the effect of over-smoothing by using increasing values of parameters in the context of gene–gene interactions and over-smoothing and how it affects the recovered biological insights.

Concerning the construction of ***M*^t^** based on a PCA-UMAP initialization with sc-PHENIX, it separates distinct phenotypes enough for a complete cell graph imputation. Our imputation approach enables sharing information between more local neighbors using higher values of parameters, avoiding over-smoothing. This allowed us to find local, global, and continuum details of cell phenotypes that otherwise non-imputation or MAGlC cannot reveal. We demonstrated with sc-PHENIX that even if we used more values of parameters (*t*, *knn*, and PCA dimensions), gene–gene interaction dynamics trends keep the same structure being more robust in that aspect than MAGIC (Appendix A). Thus, preserving the data structure is key to accurately imputing scRNA-seq data using manifold approximations. Thus, when considering the implications of sharing imputation via diffusion in a manifold, it is important to acknowledge three factors: (1) separating distinct phenotypic densities (local structure preservation), (2) arranging them well on the manifold (global structure preservation), and (3) connecting distinct phenotypes through a continuum if its valid (continuum structure preservation). Doing so increases the accuracy of the original signal, improves the detection of rare phenotypes, and enables the clear identification of the gene expression dynamics along distinct and connected cell phenotypes.

The success of sc-PHENIX in preserving data structures during imputation, compared to traditional methods that rely on 2D or 3D UMAP plots, can be attributed to our use of a higher number of UMAP components in a non-visual manner. Employing more UMAP dimensions aids in achieving a more accurate representation of high-dimensionality, which prevents data structure distortion during the optimization of the UMAP’s connectivity graph. This approach results in superior imputation performance. To corroborate this, we conducted imputations on artificially corrupted high-dimensional bulk data, incrementally increasing the number of UMAP dimensions utilized for sc-PHENIX. 

We assessed the imputation performance using the mean sample correlation (R2) between the original and imputed data. Our results indicate that adding UMAP components and stabilizing at higher dimensions improved imputation performance. These findings, detailed in Figure 2, show that using 1, 2, or 3 UMAP dimensions yielded the lowest R2 values. This discovery holds significant value for the scRNA-seq data community and the broader field of machine learning, especially in light of the criticism often directed at manifold distortions caused by 2D or 3D UMAP plots.

We performed an additional analysis, as shown in Figure 2B, which showed that sc-PHENIX with PCA-UMAP initialization is more robust against PCA dimensionality than MAGIC regarding recovering gene expression. The imputation performance of sc-PHENIX is higher when using PCA, whereas with MAGIC, its performance decreases with higher PCA dimensions. This indicates that sc-PHENIX deals much better with the detrimental effects of the curse of dimensionality generated using more PCA dimensions. Therefore, PCA components with small variances should not be underrated and discarded. The accumulation of these smallest eigenvalues can render better results to the imputation performance in combination with UMAP in the pipeline of sc-PHENIX. Additionally, we attempted to use only UMAP initialization (without PCA) for the corrupted bulk data (Figure 2D), and the imputation performance was worse than PCA-UMAP initialization but better than MAGIC. However, in the PBCM data dataset (Appendix A), UMAP initialization resulted in over-smoothed gene–gene interactions in general, with only PCA-UMAP initialization preserving gene–gene interactions through the use of imputation parameters. Also, Figure 2H demonstrated that caution is needed with variability capture practices, such as the rule of thumb of capturing at least 70% of the data’s variability. We observed that in the corrupted *C. elegans* data, using n_pca = 71 distorts the data topology and consequently over-smoothing occurs if MAGIC relies solely on this initialization. In contrast, the PCA-UMAP initialization with n_pca = 71 improved the sc-PHENIX imputation (Figure 2B). Therefore, for scRNA-seq data, PCA-UMAP initialization is necessary to preserve the data structure for sc-PHENIX imputation.

As already mentioned, UMAP has been widely used by the scRNA-seq community. However, some authors, such as Chari et al. [67], among others [11], have criticized its use because it can distort the structure of data in 2D or 3D plots, which are its typical applications. For instance, Chari et al. argue that PCA is more suitable than UMAP because UMAP does not preserve pairwise Euclidean distances from high-dimensional space. They use these distances as a metric to determine real neighbor samples and conclude that multidimensional PCA is better at preserving Euclidean distances than UMAP.

However, our work demonstrated that multidimensional PCA fails to separate different cluster samples effectively. In multidimensional PCA space, we cannot distinguish between dense regions of different classes because different types of samples still suffer from the curse of dimensionality, which leads to the concentration of distances. Thus, different cellular phenotypes will likely be near each other in high-dimensional PCA space. 

In this work using sc-PHENIX, we demonstrated that separating groups of cells that are distinct phenotypically from each other and grouping cell samples that are alike in dense regions of manifolds embedded in ***M*^t^** accurately recovers the gene expression (Figure 6 and Figure 7). This permits sharing gene expression among more cells, mitigating the risk of over-smoothing the data much better than MAGIC, revealing the underlying biological signals of scRNA-seq data. Therefore, we disagree with the conclusion of Chari et al. that Euclidean distances should always be preserved. The biology of high-dimensional sc-RNA-seq data does not always follow this assumption. If there is insufficient separation between different cell phenotype densities in high-dimensional space, methods like MAGIC or PHATE that rely on PCA may not be able to distinguish them effectively and may group them together. 

Despite the improvement of our method, it is not exempt from having deficiencies directly inherited from their original methods. For instance, UMAP has some weak points, such as the fact that using more cells can lead to a misleading optimization of low-dimensional manifolds and a loss of local structure. However, this detrimental effect can be mitigated by re-running a UMAP subsection used in a “re-clustering” analysis [68]; fewer samples facilitate embedding to better preserve the local structure. This strategy has a higher resolution among a group of similar cells [69], at least in visual projections of UMAP plots. However, the imputation based on a subsection of cells does not contemplate distinct cell phenotypes, making it inadequate to compare imputed expression cells outside the subsection. In future works, we expect that the development of manifold learning methods could better preserve the data structure for diffusion that can be used as initialization for sc-PHENIX. Finally, sc-PHENIX incorporates various UMAP variants that can be adapted to different data types. For instance, it utilizes a supervised UMAP approach for microbiome data [70] and integrates data from dual technologies such as CITE-seq through combined UMAP models. These adaptations aim to enhance the preservation of local structures. For further details on these innovative applications of UMAP variants, please refer to Appendix A.

In summary, while MAGIC is a widely recognized diffusion-based method for imputation in scRNA-seq data analysis, known for its effective manifold learning capabilities, we have identified several limitations in its application. Consequently, we propose sc-PHENIX as an enhancement to this method. We believe that sc-PHENIX’s improved data structure preservation and denoising capabilities will enable researchers to uncover new and significant insights in both biological and non-biological systems.

## Figures and Tables

**Figure 1 biology-13-00512-f001:**
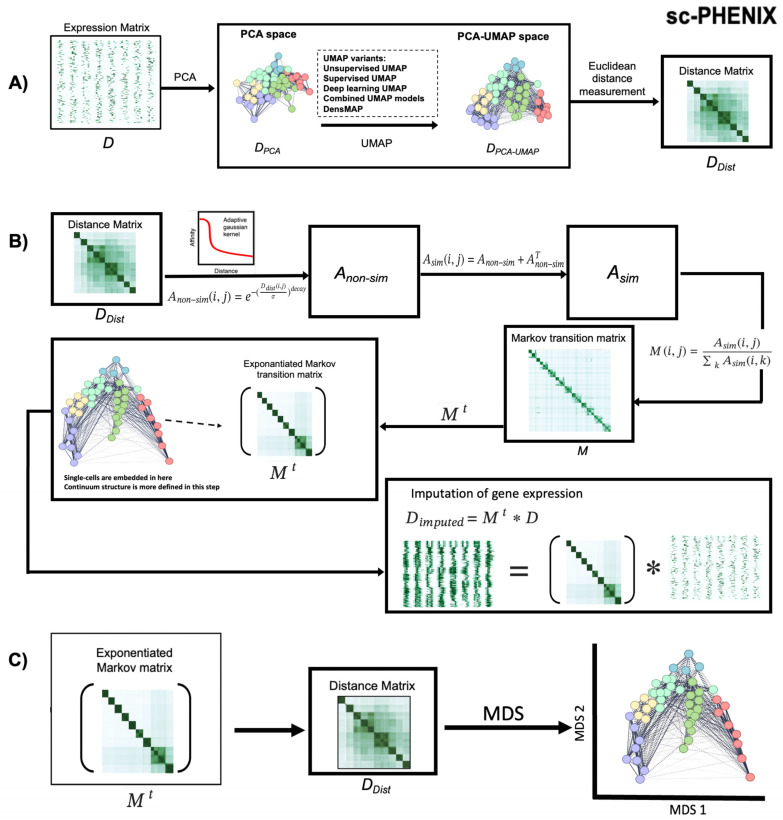
The imputation process using sc-PHENIX. The sc-PHENIX imputation approach for scRNA-seq data consists of two main steps: (**A**) The construction of the distance matrix ($D_{Dist}$): sc-PHENIX is characterized by applying PCA and then UMAP (PCA-UMAP). In this PCA-UMAP multidimensional space, sc-PHENIX constructs the best denoise representation of cell distance measurements for the diffusion process to preserve data structures. (**B**) The diffusion maps for imputation: the imputation process using diffusion maps consists of several steps: (i) Construction of the Markov transition matrix ***M*** from $D_{Dist}$: sc-PHENIX uses the adaptive Gaussian kernel to generate a non-symmetric affinity matrix ($A_{non-sim}$), it is symmetrized. Then, it is normalized to generate (***M***). (ii) Diffusion process: ***M*** is exponentiated to a chosen power *t* (random walk of length *t* named “diffusion time”) to obtain the exponentiated Markov matrix (***M^t^***). The ***M^t^*** graph well preserves the continuum structure better than the previous steps. (iii) Imputation: This step consists of multiplying the exponentiated Markov matrix (***M^t^***) times the single-cell-matrix data **D** to obtain an imputed and denoised scRNA-seq matrix ($D_{imputed}$). Note: The symbol * used in this figure indicates matrix multiplication for ***M^t^*** and ***D*** in a computational formalism, which is equivalent to the formal mathematical notation ***M^t^*** ⋅ ***D***. All equations are described in the Section 2 section. (**C**) Visualization of the exponentiated Markov matrix: We convert the ***M^t^*** into a distance matrix ($D_{Dist}$). Then, we apply a multidimensional scaling method to project data in 2D or 3D dimensions. This projection can be used as a heuristic method for quality control of imputation.

**Figure 2 biology-13-00512-f002:**
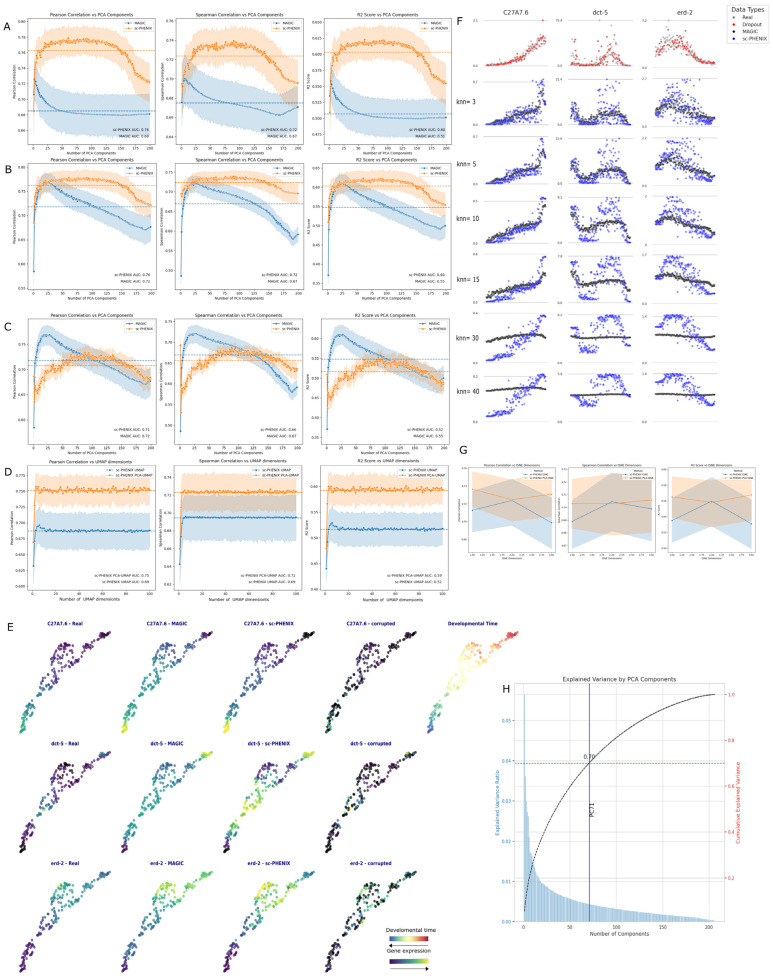
**Comparative Analysis of Imputation Methods on Corrupted Microarray Data.** We corrupted the data by randomly assigning zeros to 80% of the values and compared the imputed data with the original data. We fixed *t* = 5 and *decay* = 15. We used Pearson, Spearman, and R^2^ metrics to evaluate the imputed data compared to the original data. (**A**) Imputation based on *knn* = 30. (**B**) sc-PHENIX *knn* = 30 and MAGIC *knn* = 5. This comparison aims to identify the optimal scenarios for using MAGIC and sc-PHENIX: a low *knn* value for MAGIC and a high *knn* value for sc-PHENIX. (**C**) sc-PHENIX *knn* = 30 and MAGIC *knn* = 30. (**D**) Comparison of the performance of sc-PHENIX imputation using UMAP without initialization and with PCA-UMAP initialization. (**E**) 2D-UMAP plots of the microarray data visualizing the gene values from corrupted data, imputed data with sc-PHENIX and MAGIC, and the developmental time. (**F**) Observation of gene trends along the developmental time (left to right) with original values, imputed values, and points dropped out at 60%. (**G**) t-SNE and PCA-t-SNE initialization for sc-PHENIX on the 80% corrupted data, with the same metrics used in (**A**). (**H**). Explained Variance by PCA Components: The graph shows the explained variance ratio (blue bars) and cumulative explained variance (black dashed line) for the principal components (PCs) of the dataset. The vertical blue line indicates PC 71, where the cumulative explained variance reaches 70% (green dashed line). It demonstrates the number of components required to capture 70% of the total variance in the dataset. Note: The dotted lines represent the global mean of the metric for all samples, with orange for sc-PHENIX and blue for MAGIC.

**Figure 3 biology-13-00512-f003:**
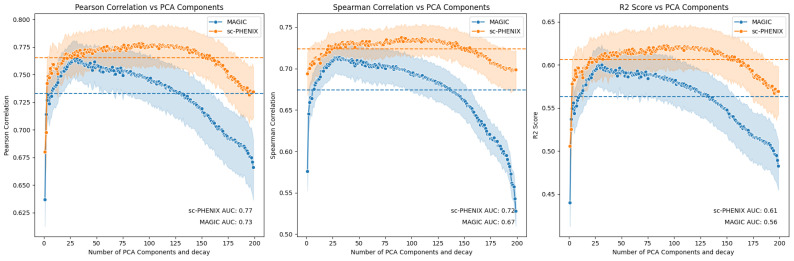
**Implications of the *Decay* Parameter and PCA Dimensionality on sc-PHENIX and MAGIC.** The parameters are the same as in Figure 2B, with the only difference being that the *decay* values increase as the PCA dimensionality increases. Additionally, we fixed *t* = 5 sc-PHENIX *knn* = 30 and MAGIC *knn* = 5. This comparison aims to identify the optimal scenarios for using MAGIC and sc-PHENIX: a low *knn* value for MAGIC and a high *knn* value for sc-PHENIX. The dotted lines represent the global mean of the metric for all samples, with orange for sc-PHENIX and blue for MAGIC.

**Figure 4 biology-13-00512-f004:**
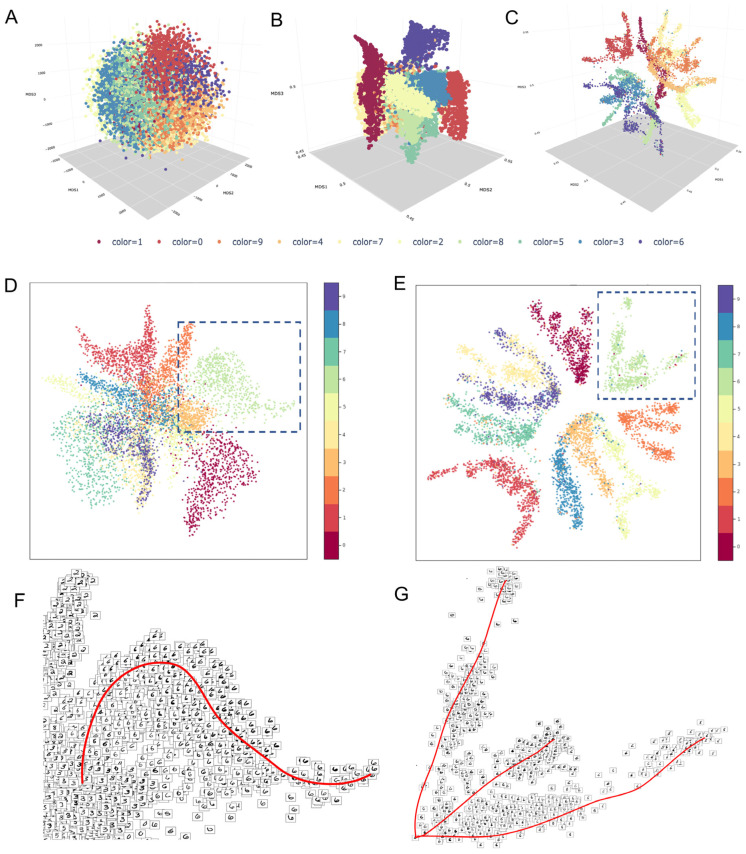
**MNIST dataset visualization: Multidimensional scaling.** (**A**). Three-dimensional MDS plot of the PCA manifold (500 PC’s, *knn* = 30 and *t* = 5). (**B**). Three-dimensional MDS plot of the exponentiated Markov matrix (500 PCs as input, *knn* = 30 and *t* = 5). (**C**). Three-dimensional MDS plot of the exponentiated Markov matrix, ***M*^t^** (500 PCs transformed into 60 UMAP components as input, *knn* = 50, *t* = 10). (**D**). Two-dimensional MDS plot of the exponentiated Markov matrix, ***M*^t^** (500 PCs as input, *knn* = 30 and *t* = 5, MNIST), as MAGIC. (**E**). Two-dimensional MDS plot of the exponentiated Markov matrix, ***M^t^*** (500PC’s transformed into 60 UMAP components as input, *knn* = 50, *t* = 10), as sc-PHENIX. (**F**). One branch of the 6’s digit images of the PCA space (subsection of Figure 2D). Redline line color indicates the branch continuum, and red lines were drawn to visualize the branches. (**G**). Three branches of the 6’s digit images subsection of the PCA-UMAP (subsection of Figure 2E).

**Figure 5 biology-13-00512-f005:**
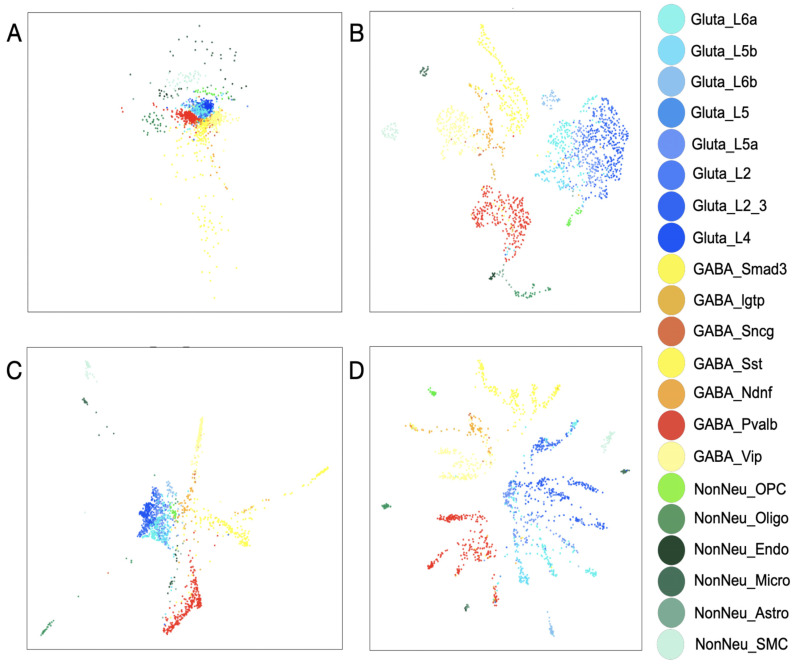
**Multidimensional scaling visualization of a neuronal scRNA-seq dataset.** (**A**) Two-dimensional MDS plot of the PCA manifold (500 PC’s). (**B**). Two-dimensional UMAP plot (2 UMAP components). (**C**) Two-dimensional MDS plot of the exponentiated Markov matrix, **M^t^** (500 PC’s, *knn* = 30, *t* = 5). (**D**). Two-dimensional MDS plot of the exponentiated Markov matrix, **M^t^** (500 PCs transformed into 60 UMAP components as input, *knn* = 30 and, *t* = 5) of the adult mouse visual cortex cells dataset. Note: For the adult mouse visual cortex cells dataset, three main clusters are GABAergic (red-yellowish), glutamatergic (blueish), and non-neuronal (greenish) cell types.

**Figure 6 biology-13-00512-f006:**
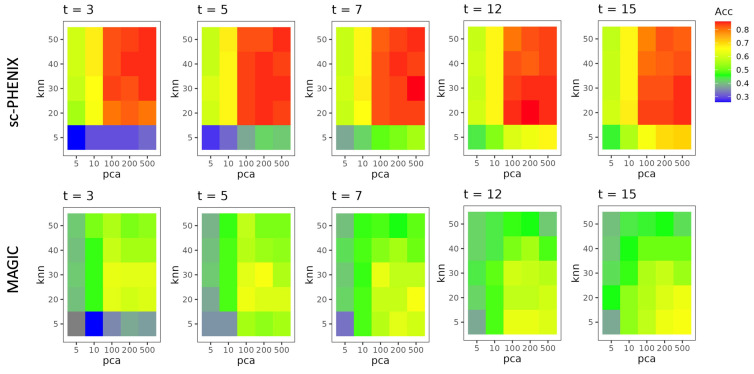
Overview of the accuracy of the *knn* classifier model on the 2D-MDS space derived from MAGIC’s and sc-PHENIX’s exponentiated Markovian matrix. Each plot represents different combinations of diffusion parameters *t*, *knn* values, and PCA dimensions. The color gradient, using a rainbow palette, indicates the classifier accuracy, with red representing higher accuracy and blue representing lower accuracy. The top row corresponds to MAGIC, while the bottom row corresponds to sc-PHENIX. The results show how increasing *t*, *knn*, and PCA dimensions affect local structure preservation.

**Figure 7 biology-13-00512-f007:**
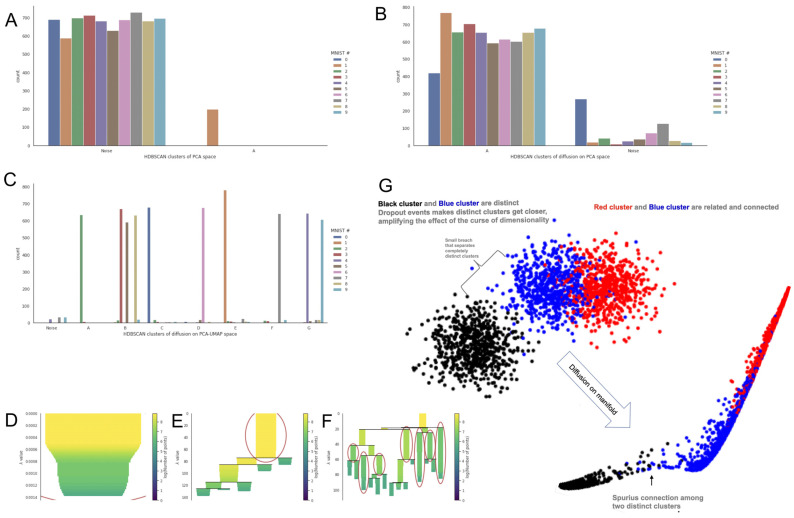
**HDBSCAN clusters on the exponentiated Markov matrix of sc-PHENIX.** Clusters were assigned as letters (A, B, C, etc.). (**A**) MNIST samples distribution of different HDBSCAN clusters (PCA space). (**B**) Distribution of MNIST samples on different HDBSCAN clusters of the **M^t^** (diffusion on PCA space, also known as MAGIC). (**C**) Distribution of MNIST samples on different HDBSCAN clusters of the **M^t^** (diffusion on PCA-UMAP space, also known as sc-PHENIX). (**D**) Condense tree plot (PCA space). (**E**) Condense tree plot (diffusion on PCA space). (**F**) Condense tree plot (diffusion on PCA-UMAP space). (**G**) Scheme of an inaccurate diffusion process. Diffusion in PCA space connects two distinct clusters (black and blue). This connection occurs in the proximate regions between different clusters (distinct cell phenotypes) separated by a small gap. Due to the diffusion process, this artifact includes spurious neighboring samples that do not share similar features. This occurs because all points (cells) are relatively close to each other in the multidimensional PCA space, and PCA does not provide sufficient separation. Note: In (**D**–**F**), the red circles indicate the most stable and persistent clusters identified by HDBSCAN. These clusters are highlighted because they exhibit higher stability, measured by the λ values at which points remain within them before splitting into smaller clusters, indicating their significance and robustness.

**Figure 8 biology-13-00512-f008:**
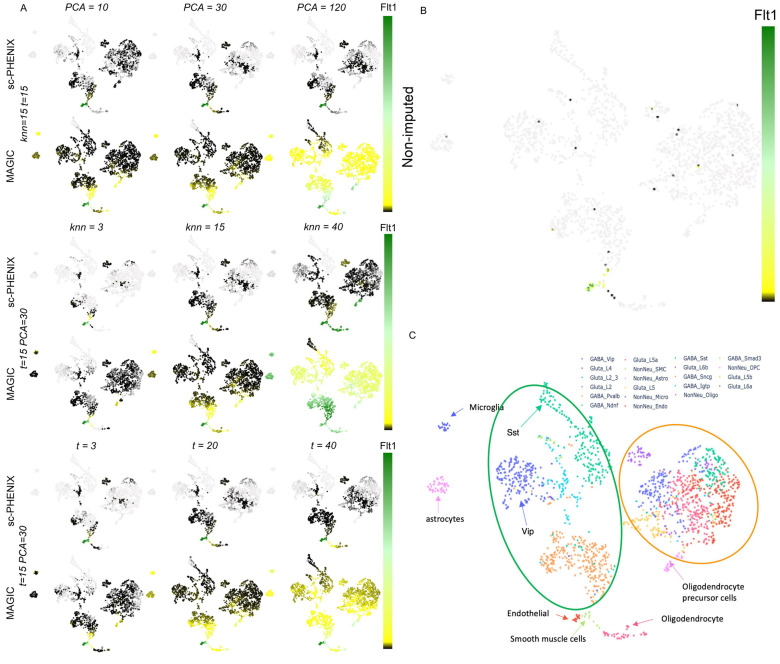
**Imputation of the adult mouse visual cortex using MAGIC and sc-PHENIX.** (**A**) Recovered Flt1 expression by MAGIC and sc-PHENIX are visualized on UMAP projection of the adult mouse visual cortex cells dataset with different parameter combinations. (**B**) The non-imputed expression values of Flt1 are visualized on UMAP projection of the adult mouse visual cortex cells dataset. (**C**) In the 2D UMAP projection of the adult mouse visual cortex cells dataset (without imputation), three main clusters are GABAergic (green circle), glutamatergic (orange circle), and non-neuronal cell types, with 21 cell phenotypes in total. Different parameters were used to see the effect of over-smoothing for MAGIC and sc-PHENIX methods.

**Figure 9 biology-13-00512-f009:**
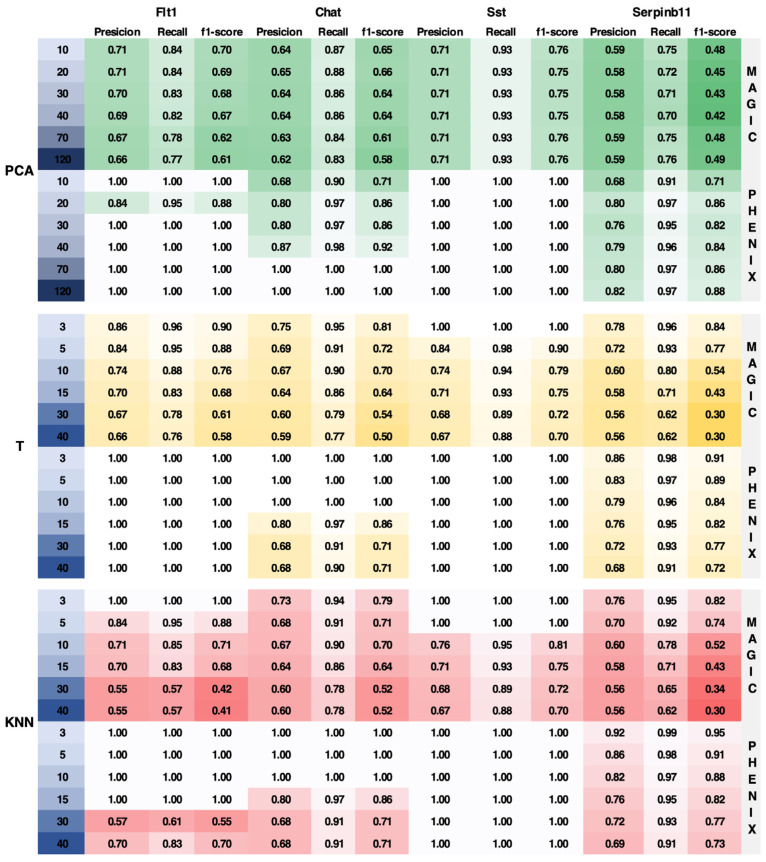
**Imputation performance of MAGIC and sc-PHENIX imputation through different increasing combinations of parameters.** Here, we show the precision, recall, and f1-score performance metrics for imputation using different combinations of parameters of *knn*, *t*, and PCA dimensions. We used *Flt1* for NonNeu_Endo and NonNeu_SMC cell types, *Chat* for GABA_Vip cell type, *Sst* for GABA_Sst cell type, and *Serpin11* for Gluta_L6B cell types. The differential expression for the imputed gene markers was set to Fold Change = 2.0 and FWRD = 0.05 using Tukey’s HSD (honestly significant difference). Note: To increase values for PCA, we set it to *knn* = 15 and *t*= 15. To increase values for *t*, we set it to *knn* = 15 and *n_pca* = 30. For increasing values of *knn*, we set *t*= 15 and PCA= 30, similar to that shown in Figure 6 and Appendix A. Note: Gene markers: *Flt1* (vascular endothelial growth factor receptor 1), and *Chat* (Choline Acetyltransferase), *Sst* (Somatostatin), and *Serpinb11* (Serpin Family B Member 11).

**Figure 10 biology-13-00512-f010:**
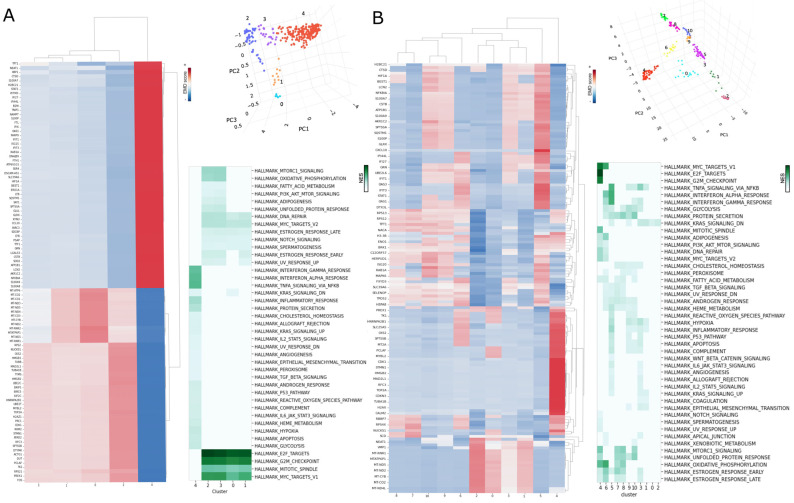
**Characterization of MCF7 MCTS polarization phenotypes based on the recovered data by MAGIC and sc-PHENIX.** (**A**) Downstream analysis post imputation with MAGIC (diffusion on PCA). (**B**) Downstream analysis post imputation with sc-PHENIX (diffusion on PCA-UMAP). The downstream analysis is: (i) Differential expression by cluster, see heatmap of DEG (Differential expressed genes) using EMD score (red and blueish colored), DEG (rows) for each HDBSCAN cluster (columns in numbers). (ii) GSEA heatmap (greenish colored), the statistical significance is pronounced at FDR < 0.05, the normalized enrichment score (NES), only positive enrichment values were considered for visualization. (iii) Only dense cluster clusters are visualized in the 3D-PCA space of the recovered (imputed) data. Note: the 3D PCA plots here come from the interactive plots in Appendix A (MAGIC) and Appendix A (sc-PHENIX) in order to see more details in a 3D plot context.

**Figure 11 biology-13-00512-f011:**
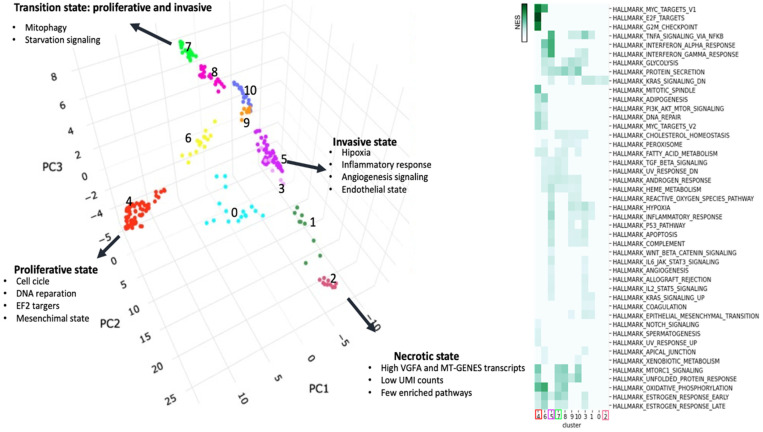
**Overview of the characterization of extreme and transition state cell phenotypes from sc-PHENIX.** Overview of the characterization of cell phenotypes obtained from the imputed data with sc-PHENIX. Arrows indicate clusters that are in the extreme of clusters (archetypes) or extreme phenotypes in cell space in the 3D-PCA plot, and it is already known that extreme archetypes are samples that differ in gene expression. On the right: GSEA heatmap (greenish colored), the statistical significance is pronounced at FDR < 0.05, the normalized enrichment score (NES), only positive enrichment values were considered for visualization. There are 4 extreme clusters that we named (1) Proliferative state, where enriched pathways are related to a proliferation state. (2) Invasive state, where enriched pathways are related to a hypoxic phenotype in which inflammatory responses and angiogenesis pathways are present. (3) Necrotic state, where we already identified over-expression of VEGFA and MT GENES, low UMI counts, and diminished pathways. (4) Transition state, which shares HALLMARKS from the extreme clusters 4 and 5 but based on the REACTOME database (Appendix A), this cluster presents high values of NES of the mitophagy and starving signaling pathways in the extreme clusters 7 and 8.

## Data Availability

Publicly available datasets were analyzed in this study. sc-PHENIX is open-source and available at https://github.com/resendislab/sc-PHENIX, accessed on 23 February 2024.

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
