# Peer review of "Diffusion on PCA-UMAP Manifold: The Impact of Data Structure Preservation to Denoise High-Dimensional Single-Cell RNA Sequencing Data"

_biology, 2024, doi:10.3390/biology13070512_

Round 1

Reviewer 1 Report (Previous Reviewer 1)

Comments and Suggestions for Authors

The author addressed all the questions.

Reviewer 2 Report (Previous Reviewer 2)

Comments and Suggestions for Authors

The authors have addressed all of our recommendations. We accept the manuscript in current form.

This manuscript is a resubmission of an earlier submission. The following is a list of the peer review reports and author responses from that submission.

Round 1

Reviewer 1 Report

Comments and Suggestions for Authors

Christian Padron-Manrique et al. have developed sc-PHENIX, an imputation method for single-cell RNA sequencing data, designed to minimize noise. This method, building upon the foundational concepts of MAGIC, constructing a cell neighborhood graph by PCA-UMAP to enhance the imputation of gene expression among similar cells. sc-PHENIX surpasses MAGIC and other state-of-the-art approaches and promises to be valuable for the research community. I have some suggestions:

1.        In Figure 4, the authors present two-dimensional MDS plots of the adult mouse visual cortex under varying parameters for kNN, t, and PCA dimensions. They argue that PCA-UMAP is superior to PCA alone, as it better preserves data structure. However, instead of solely relying on 2D MDS visualizations, it would be beneficial for the authors to provide numerical metrics to quantify the preservation of data structure under these conditions.

2. In Figure 4, the authors utilize a t-value greater than 15 for most comparison cases. However, the default parameter for MAGIC is t=3. Could a t-value of 15 be considered an excessively large parameter for MAGIC? This consideration should be mentioned in the manuscript.

3.     In Figure 8, the authors compare the imputed data matrices of the MCTS dataset using sc-PHENIX and MAGIC through 3D-PCA maps, where they observed more trajectories in the datasets smoothed by sc-PHENIX. However, UMAP and t-SNE are more commonly employed for visualizing single-cell data in recent literature. I suggest the authors also include UMAP and t-SNE results in the supplementary figures to demonstrate the differences between sc-PHENIX and MAGIC. Additionally, since imputation results are significantly influenced by parameter choices, the authors should provide details of the smoothing parameters used by sc-PHENIX and MAGIC in Figure 8.

Reviewer 2 Report

Comments and Suggestions for Authors

Comments on the Quality of English Language

 Extensive editing of English language required. There are several sentences which have to be corrected and improved throughout the manuscript
